# A Msp1-containing complex removes orphaned proteins in the mitochondrial outer membrane of *T. brucei*

Markus Gerber[1], Ida Suppanz[2], Silke Oeljeklaus[3] ®, Moritz Niemann[1] ®, Sandro Käser[1], Bettina Warscheid[2,3] ®, André Schneider[1,4] ®, Caroline E Dewar[1] ®

The AAA-ATPase Msp1 extracts mislocalised outer membrane proteins and thus contributes to mitochondrial proteostasis. Using pulldown experiments, we show that trypanosomal Msp1 localises to both glycosomes and the mitochondrial outer membrane, where it forms a complex with four outer membrane proteins. The trypanosome-specific pATOM36 mediates complex assembly of α-helically anchored mitochondrial outer membrane proteins such as protein translocase subunits. Inhibition of their assembly triggers a pathway that results in the proteasomal digestion of unassembled substrates. Using inducible single, double, and triple RNAi cell lines combined with proteomic analyses, we demonstrate that not only Msp1 but also the trypanosomal homolog of the AAA-ATPase VCP are implicated in this quality control pathway. Moreover, in the absence of VCP three out of the four Msp1-interacting mitochondrial proteins are required for efficient proteasomal digestion of pATOM36 substrates, suggesting they act in concert with Msp1. pATOM36 is a functional analog of the yeast mitochondrial import complex complex and possibly of human mitochondrial animal-specific carrier homolog 2, suggesting that similar mitochondrial quality control pathways linked to Msp1 might also exist in yeast and humans.

## Introduction

The mitochondrial outer membrane (OM) forms the interface between mitochondria and the cytosol, and many of its proteins have important functions in cytoplasmic-mitochondrial communication. Integral OM proteins are often α-helically anchored, and many have just a single transmembrane domain (TMD). Intriguingly, there are at least three unrelated protein factors mediating the biogenesis of α-helically anchored OM proteins in different eukaryotic clades.

The mitochondrial import complex (MIM) was discovered in *Saccharomyces cerevisiae* and consists of two small proteins, Mim1 and Mim2, that are restricted to fungi (Becker et al, 2011; Papić et al, 2011; Dimmer et al, 2012; Doan et al, 2020). In the parasitic protozoan *Trypanosoma brucei*, the kinetoplastid-specific peripheral atypical protein translocase of the outer membrane 36 (pATOM36) has the same function (Käser et al, 2016; Bruggisser et al, 2017). Expression of pATOM36 in yeast lacking the MIM complex restores growth under non-permissive conditions, and vice versa, expression of the MIM complex complements the OM protein biogenesis defect in pATOM36-ablated trypanosomes (Vitali et al, 2018). In human cells, the mitochondrial animal-specific carrier homolog 2 (MTCH2) is necessary and sufficient to insert α-helically anchored membrane proteins into the OM (Guna et al, 2022). However, at least in yeast, spontaneous insertion into the OM also seems possible for some proteins (Kemper et al, 2008; Vögtle et al, 2015).

Safeguarding mitochondrial functions requires mitochondria-associated degradation (MAD) pathways that survey the OM and guarantee that its proteins are correctly targeted and assembled (Mohanraj et al, 2020; den Brave et al, 2021; Krämer et al, 2021). The highly conserved Msp1, an ATPase associated with diverse cellular activities (AAA), plays a key role in this process. It consists of an N-terminal TMD and a C-terminal AAA domain, which faces the cytosol (Nakai et al, 1993), and localises to both the OM and the peroxisomal membrane. Msp1 extracts mislocalised and/or misassembled proteins from the OM and feeds them to the cytosolic proteasome (Chen et al, 2014; Okreglak & Walter, 2014; Weir et al, 2017; Wohlever et al, 2017; Weidberg & Amon, 2018). Non-mitochondrial tail-anchored (TA) proteins, which have a single TMD at their C-terminus, can be prone to OM mistargeting under both normal and stress conditions (Kalbfleisch et al, 2007; Chen et al, 2014; Okreglak & Walter, 2014; Rao et al, 2016; Costello et al, 2017; Weir et al, 2017; Wohlever et al, 2017). The latter includes a deficient guided-entry of TA protein pathway in the ER or an impaired peroxisomal targeting machinery (Schuldiner et al, 2008; Jonikas et al, 2009; Chen et al, 2014; Okreglak & Walter, 2014). There is no

[1]Department of Chemistry, Biochemistry and Pharmaceutical Sciences, University of Bern, Bern, Switzerland    [2]Signalling Research Centres BIOSS and CIBSS, University of Freiburg, Freiburg, Germany    [3]Faculty of Chemistry and Pharmacy, Biochemistry II, Theodor Boveri-Institute, University of Würzburg, Würzburg, Germany    [4]Institute for Advanced Study (Wissenschaftskolleg) Berlin, Berlin, Germany

Correspondence: andre.schneider@unibe.ch; bettina.warscheid@uni-wuerzburg.de; c.dewar1@lancaster.ac.uk
Ida Suppanz's present address is Max-Planck-Institute of Immunobiology and Epigenetics, Freiburg, Germany
Moritz Niemann's present address is Mattei team, EMBL Imaging Centre, Heidelberg, Germany
Sandro Käser's present address is Institute of Cell Biology, University of Bern, Bern, Switzerland

clear sequence consensus between Msp1 substrates (Chen et al, 2014; Okreglak & Walter, 2014; Weir et al, 2017; Wohlever et al, 2017; Dederer et al, 2019; Li et al, 2019). However, Msp1 recognises and extracts orphan TA proteins that are normally found in a complex, which suggests that their oligomeric state is an important determinant (Weir et al, 2017; Dederer et al, 2019). Intriguingly, mitochondrial Msp1 is not known to form stable complexes with other proteins and appears to extract its substrates from the OM without help from other proteins (Wohlever et al, 2017; Dederer et al, 2019). However, adaptor proteins may still be required for substrate selectivity or regulation of activity. For example, Msp1 is able to clear stuck precursor proteins from the TOM complex via a transient interaction with the inducible peripheral OM protein Cis1 and the TOM receptor Tom70 in response to mitochondrial protein import stress (Weidberg & Amon, 2018).

Msp1 deletion in yeast causes a mild growth phenotype only, which suggests some redundancy in OM quality control (Chen et al, 2014; Okreglak & Walter, 2014). In line with this, it was shown that under stress conditions, the AAA-ATPase VCP, a soluble cytoplasmic component of the ER-associated protein degradation system, can also extract mistargeted proteins from the OM (Heo et al, 2010). For degradation by the proteasome, proteins generally require ubiquitination. It has been shown that mislocalised proteins can be extracted from the OM by Msp1 and transferred to the ER, where they are ubiquitinated by the ER-resident E3 ligase Doa10. This allows for their extraction from the membrane by VCP and subsequent degradation by the proteasome (Dederer et al, 2019; Matsumoto et al, 2019). However, as E3 ligases normally have specific sets of substrates, this pathway might not be required for all Msp1 substrates, and some may be degraded by the proteasome without prior ubiquitination (Matsumoto et al, 2019).

Studies of mitochondrial processes, including MAD pathways, have mainly focused on yeast and mammals, which belong to the same eukaryotic supergroup of the Opisthokonts. However, a better understanding of their basic features and evolutionary history requires that these processes be studied across divergent eukaryotes. Arguably the best-studied mitochondrion outside of yeast and mammals is that of *T. brucei*. It belongs to the Discoba supergroup, which is essentially unrelated to the Opisthokonts (Verner et al, 2015; Harsman & Schneider, 2017; Schneider, 2020).

It has previously been shown that ablation of pATOM36 triggers a MAD pathway, resulting in the proteasomal digestion of destabilised pATOM36 substrates from the OM. Results of the present study, using cells depleted for Msp1 and/or TbVCP, are consistent with the notion that TbVCP and TbMsp1 contribute to this pathway. In addition, we found four integral OM proteins that interact with TbMsp1 and showed that ablation of three of them interferes with the MAD pathway in cells where TbMsp1 levels are not affected.

## Results

### TbMsp1 interacts with proteins of the mitochondrial OM and the glycosomes

Msp1 is highly conserved within eukaryotes, with TbMsp1 showing 34.5 and 33.5% identity to that of yeast and human Msp1, respectively. This

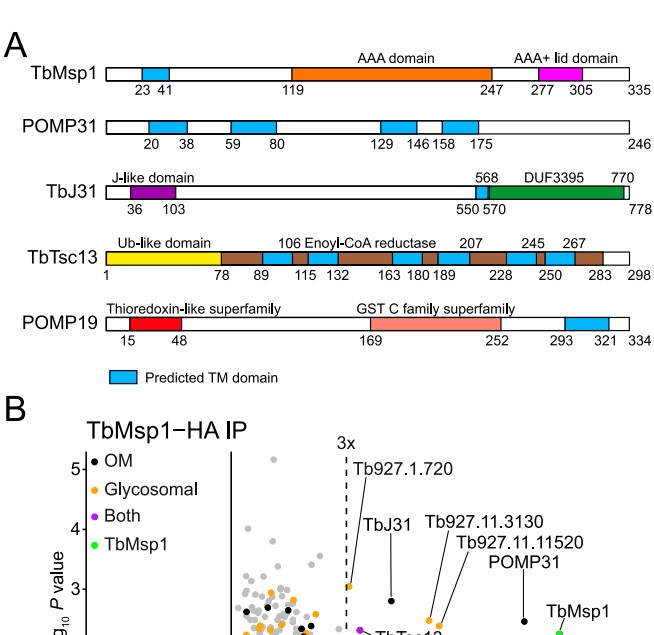

**Figure 1.  TbMsp1 forms complexes in the OM and glycosomes.**
**(A)** Schematic depiction of predicted domain structures of TbMsp1 and four interacting OM proteins. The indicated domains were predicted as described in the Material and Methods section. **(B)** TbMsp1 complexes were immunoprecipitated from crude mitochondrial fractions of differentially stable isotope labelling by amino acids in cell-labelled 29.13 parent cells and cells expressing in situ tagged TbMsp1-HA analysed by quantitative mass spectrometry (n = 3). Proteins found to be significantly enriched more than threefold in TbMsp1 complexes are labelled with either their name or their accession number.

conservation is in contrast to many other trypanosomal OM proteins, most of which are specific to kinetoplastids (Niemann et al, 2013). TbMsp1 has the expected conserved sequence motifs including the AAA domain and the Walker A and Walker B motifs required for ATP binding and hydrolysis (Figs 1A and S1). To identify TbMsp1 interaction partners and determine its intracellular localisation, we produced a cell line expressing a C-terminally in situ HA-tagged TbMsp1 variant. Digitonin-extracted crude mitochondrial fractions of this cell line were subjected to a stable isotope labelling by amino acids in cell culture (SILAC) immunoprecipitation experiment using anti-HA antibodies. TbMsp1-HA precipitated 10 proteins with enrichment factors of more than threefold (Fig 1B). From previous proteomic analyses, three were identified as OM proteins, five are glycosomal proteins, TbTsc13 showed both localisations, and Tb927.3.4500 is the cytosolic fumarate hydratase, which was hypothesised to interact with the cytosolic side of the glycosomal membrane (Colasante et al, 2006; Coustou et al, 2006; Niemann et al, 2013; Güther et al, 2014). Of the glycosomal proteins, the peroxisome biogenesis protein Pex11 (Tb927.11.11520), tyrosine phosphatase (Tb927.10.10610), glycosomal metabolite transporters GAT1 (Tb927.4.4050), and GAT2 (Tb927.11.3130) all contain TMDs (Lorenz et al, 1998; Yernaux et al, 2006; Igoillo-Esteve et al, 2011),

whereas phosphoglycerate kinase A (Tb927.1.720) is localised in the glycosomal lumen (Alexander & Parsons, 1993; Peterson et al, 1997).

In the present study, we focused on the four most enriched OM proteins (Fig 1). The first is the protein of the mitochondrial OM proteome 31 (POMP31), which is a kinetoplastid-specific protein of unknown function with four predicted TMDs. The second one is TbJ31, a J-like protein which has a single predicted TMD (Bentley et al, 2019). It is the homolog of mammalian DNAJC11, with which it also shares the domain of unknown function 3,395 (Muñoz-Gómez et al, 2015). The third is POMP19, a kinetoplastid-specific protein with a single TMD that contains a predicted thioredoxin-like and a predicted glutathione S-transferase domain, and the fourth is TbTsc13, which was previously detected in a proteomic study of glycosomes (Güther et al, 2014). It shows homology to the mammalian enoyl-CoA reductase of the ER elongase complex and has six predicted TMDs (Cinti et al, 1992). TbTsc13 contains a predicted ubiquitin-like domain at the N-terminus.

Cell fractionation using low concentration of digitonin results in a soluble fraction, containing the cytosol, and a crude mitochondrial fraction which also contains most of the ER marker binding protein (BiP), the glycosomal marker aldolase (ALD), and other particulate cell components (Fig S2A and B). TbMsp1-HA and its four epitope-tagged OM interactors co-fractionated with the voltage-dependent anion channel (VDAC), as would be expected for mitochondrial proteins (Fig S2A). A proteinase K protection assay furthermore showed that the mitochondria were still intact in the digitonin pellet because the intermembrane space-localised Tim9 and the matrix marker mitochondrial heat shock protein 70 (mHsp70) were protected from the added proteinase K and were only digested after the addition of Triton X-100 (Fig S2C). TbMsp1 and three interactors, on the other hand, were as proteinase K-sensitive as the atypical protein translocase of the OM 69 (ATOM69), the OM protein that serves as a control (Fig S2C). Finally, TbMsp1 and its interactors were predominantly recovered in the pellet when subjected to alkaline carbonate extraction at high pH, indicating that, in line with their predicted TMDs, they are all integral membrane proteins (Fig S2A, lower panels).

Immunofluorescence of cells expressing either TbMsp1-myc or an epitope-tagged interactor revealed a close degree of co-localisation of POMP31, TbJ31, and POMP19 with the mitochondrial marker ATOM40 (Fig S3A). As expected from the SILAC-pulldown experiment (Fig 1B) and previous analyses (Cinti et al, 1992; Güther et al, 2014), TbMsp1-myc and TbTsc13-HA are not exclusively mitochondrially localised. TbMsp1-myc, in addition to mitochondrial staining, partially co-localised with the glycosomal marker ALD (Fig S3B). The localisation of TbTsc13-HA, in line with its predicted function as an enoyl-CoA reductase, partially overlapped with the ER luminal BiP (Fig S3C).

In addition, normalised abundance profiles of untagged native TbMsp1 and its four interactors from a previous proteomic analysis with six subcellular fractions, including crude and pure OM, confirm the OM localisation of all four proteins (Fig S4) (Niemann et al, 2013).

Finally, we validated the interactions of the four proteins with TbMsp1 and between each other by immunoprecipitations using cell lines in which both Msp1 and one candidate interactor were epitope-tagged. It is important to note that expression of the tagged TbMsp1 only marginally affects growth (Fig S5). Interactions could

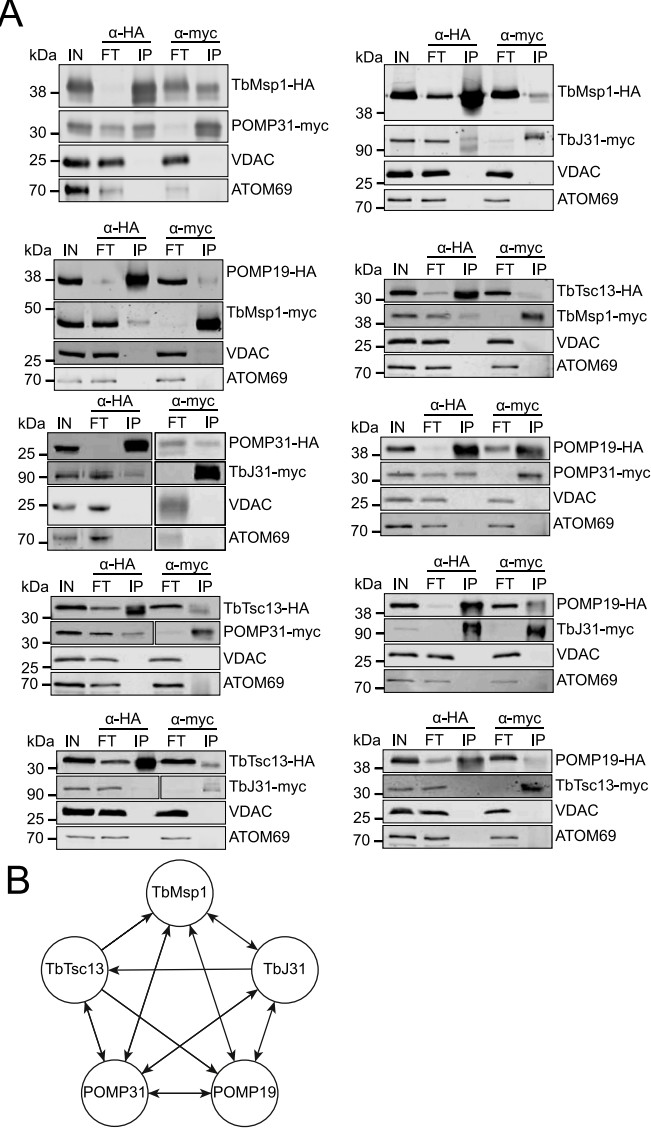

**Figure 2. Reciprocal IPs reveal a TbMsp1-centric interaction network in the OM.**
**(A)** Crude mitochondrial fractions from cells overexpressing the indicated C-terminally myc- and HA-tagged proteins were analysed by immunoprecipitation. Crude mitochondrial fractions (IN), unbound proteins (FT), and final eluates (IP) were separated by SDS–PAGE. Resulting immunoblots were probed with anti-tag antibodies and antisera against voltage-dependent anion channel and ATOM69. **(B)** Summary of the confirmed interactions detected by coimmunoprecipitation. Two-sided arrows indicate reciprocal interactions.

be confirmed between TbMsp1 and each of POMP31, TbJ31, POMP19, and TbTsc13 (Fig 2A), whereas interactions were not detected between these proteins and the most abundant OM protein, VDAC, or the α-helically anchored protein import receptor ATOM69. Using the same method, we could also detect mostly reciprocal interactions between POMP31, TbJ31, POMP19, and TbTsc13. As a further control, we subjected cell lines individually expressing tagged TbMsp1 and each of its four tagged interactors to pulldown with anti-HA and myc beads, respectively. As expected, the tagged proteins were only recovered in the pellet when using resin with matching anti-HA or myc beads; no unspecific interaction of the tagged proteins with the

## A

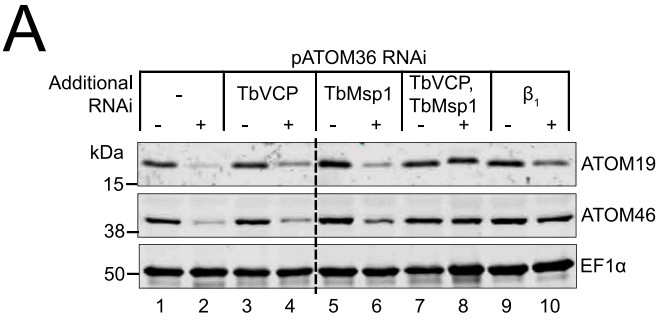

## B

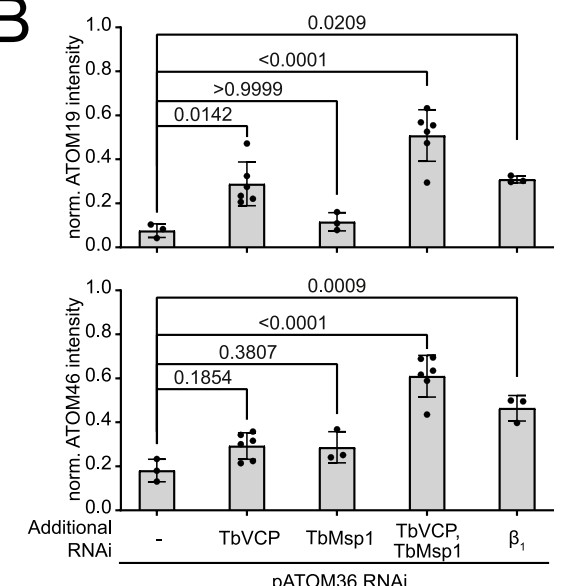

**Figure 3. TbVCP and TbMsp1 are synergistically involved in the degradation of ATOM19 and ATOM46 by the cytosolic proteasome.**
**(A)** Western blot analysis of total cellular extract (3 × 10⁶ cells each) of the indicated uninduced and induced single, double, and triple RNAi cell lines (−/+ Tet), probed with ATOM19 and ATOM46 antisera. EF1α serves as loading control. **(B)** Quantifications of ATOM46 and ATOM19 levels in the RNAi cell lines from immunoblots shown in (A). The signal for each sample was normalised to its respective EF1α signal and then to the respective signal in uninduced cells. Data are presented as mean values with error bars corresponding to the SD (n = 3–6). The P-values indicated in the graph were calculated using a one-way ANOVA followed by a Bonferroni post hoc test to allow for multiple comparisons.

resin was observed (Fig S6). In summary, these results suggest that at least a fraction of all five proteins are present in the same protein complex (Fig 2B).

Finally, to investigate the importance of TbMsp1 and the four TbMsp1-interacting proteins for cell viability, we produced inducible RNAi cell lines targeting the ORFs of these proteins. However, despite the fact that the RNAi efficiently depleted the corresponding target mRNAs (Fig S7), only the RNAi cell line targeting TbTsc13 showed a clear inhibition of growth (Fig S7, bottom panel). This was expected as TbTsc13 is likely to play an essential role in fatty acid elongation, as in yeast (Kohlwein et al, 2001). Thus, within the limit of the RNAi analysis, which does not completely deplete gene products, TbMsp1, POMP19, POMP31, and TbJ31 are not essential for normal cell growth in the procyclic form of trypanosomes.

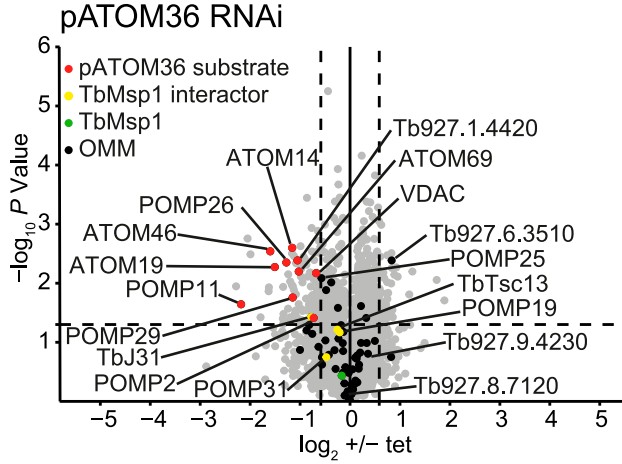

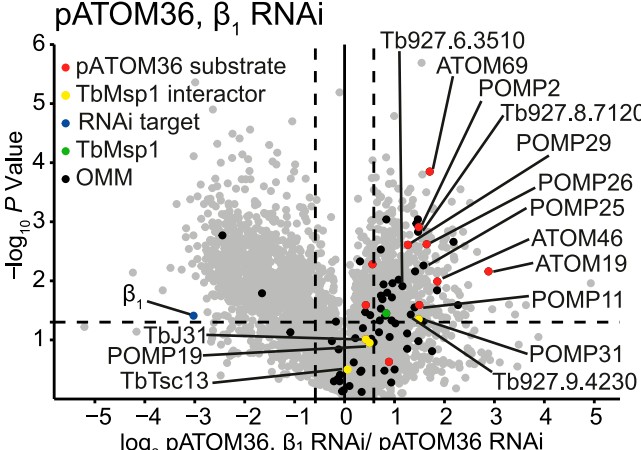

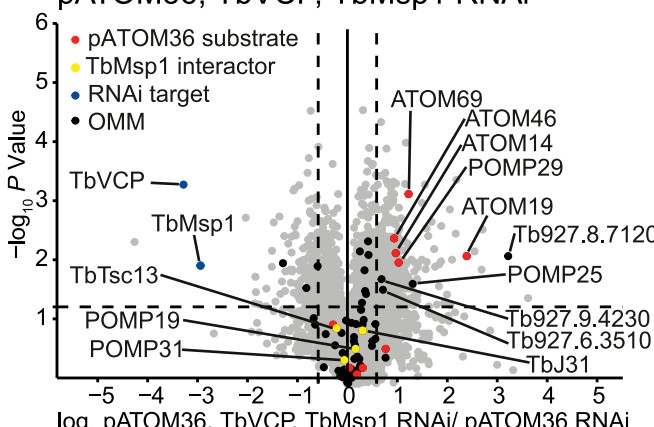

**Figure 4. Proteomic analysis shows that TbVCP and TbMsp1 are involved in the proteasomal degradation of pATOM36 substrates.**
Volcano plots visualising quantitative MS data of whole cell extracts from the indicated RNAi cell lines (n = 3) used in Fig 3. Relative protein quantification was based on peptide stable isotope dimethyl labelling. Shown are comparisons of uninduced and induced pATOM36 RNAi cells (top), induced pATOM36 RNAi cells and induced pATOM36/proteasome subunit β1 double RNAi cells (middle), and induced pATOM36 RNAi cells and induced pATOM36/TbVCP/TbMsp1 triple RNAi cells (bottom).

### pATOM36 RNAi results in proteasomal depletion of its substrates

The biogenesis of many α-helically membrane-anchored mitochondrial OM proteins is mediated by distinct protein factors in yeast (MIM complex), humans (MTCH2), and trypanosomes (pATOM36) (Becker et al, 2011; Papić et al, 2011; Dimmer et al, 2012; Käser et al, 2016; Bruggisser et al, 2017; Doan et al, 2020; Guna et al, 2022). Moreover, for the MIM complex and pATOM36, reciprocal complementation experiments demonstrate that they are functionally interchangeable (Vitali et al, 2018). In the present study, we focussed on the proteomic consequences of pATOM36 depletion in trypanosomes. The total cellular levels of the ATOM complex subunits ATOM19 and ATOM46 were massively reduced after induction of pATOM36 RNAi (Fig 3A, lanes 1 and 2; Fig 3B), in agreement with a previous proteomic analysis of crude mitochondrial fractions of pATOM36-depleted cells (Käser et al, 2016). This was confirmed when whole cell samples of the same uninduced and induced pATOM36 RNAi cell line were compared using a proteomic analysis (Fig 4, top panel). The experiment also showed that the levels of 11 OM proteins, including ATOM19 and ATOM46, were significantly reduced by more than 1.5-fold in the induced RNAi cells (Fig 4, top panel, pATOM36 substrates). This group of proteins consists of ATOM subunits, OM membrane proteins of unknown function termed POMPs (Niemann et al, 2013), TbJ31, VDAC, and the putative ABC transporter Tb927.1.4420. Eight of them have been identified as pATOM36 substrates in a previous study (Käser et al, 2016). Moreover, Tb927.1.4420 and POMP33 were found to be depleted ~1.4-fold in the previous study which is only marginally below the threshold of 1.5-fold. Approximately two-thirds of the other proteins found to be more than 1.5-fold depleted (Fig 4, top panel) belong to the mitochondrial importome, and thus their depletion is likely an indirect consequence of reduced import because of the diminished levels of the ATOM subunits. However, whereas the level of the Msp1 interactor TbJ31 was significantly decreased by 1.7-fold, the same was not the case for TbMsp1 itself or for any of the three remaining interactors. The fact that many more non-OM proteins were detected in the present experiment compared with the previous study (Käser et al, 2016) can be explained because induction of pATOM36 RNAi was 1 d longer and because, instead of crude mitochondrial fractions, whole cellular extracts were analysed.

To test whether destabilised pATOM36 substrates are digested by the cytosolic proteasome, we produced a cell line able to knock-down both pATOM36 and the proteasomal subunit $\beta 1$ (for a characterisation of all double and triple RNAi cell lines used in this study; see Fig S8). A comparison of this pATOM36/subunit $\beta_1$ double RNAi cell line with the single pATOM36 RNAi cell line by immunoblot analysis indicated that the levels of ATOM46 and ATOM19 were significantly stabilised (Fig 3A, compare lanes 2 and 10). Three-to-fivefold more ATOM19 and ATOM46 were found in cells depleted for pATOM36 and proteasomal subunit $\beta_1$ in comparison with cells only depleted for pATOM36. This was in line with data from a quantitative proteomics analysis of induced samples of the same two cell lines (Fig 4, middle panel), which showed a significant more than 1.5-fold enrichment of seven pATOM36 substrates, including ATOM19 and ATOM46, indicating that their levels were stabilised. Moreover, a number of other OM proteins not previously shown to be substrates

of pATOM36 were also stabilised. We conclude from this experiment that pATOM36 depletion triggers a pathway which feeds destabilised pATOM36 substrates to the cytosolic proteasome.

### TbMsp1 and TbVCP are implicated in proteasomal degradation of pATOM36 substrates

How can the cytosolic proteasome access membrane-integral pATOM36 substrates? In Opisthokonts, the AAA-ATPase Msp1 is able to extract TA proteins from the OM (Zheng et al, 2019). Thus, we decided to test whether TbMsp1 could be involved in the degradation of the integral OM proteins ATOM19 and ATOM46 in pATOM36-depleted cells using the same approach that was used to show the involvement of the proteasome. However, in contrast to the pATOM36/subunit $\beta_1$ double RNAi cell line (Fig 3A, compare lanes 9 and 10), combining TbMsp1 RNAi with pATOM36 RNAi (Fig 3A, compare lanes 5 and 6) did not significantly prevent the degradation of ATOM19 and ATOM46.

In Opisthokonts, the AAA-ATPase VCP is involved in various pathways that remove OM proteins from their membrane to allow for their degradation (Zheng et al, 2019). To find out whether TbVCP, the trypanosomal VCP homolog (Roggy & Bangs, 1999; Lamb et al, 2001), plays a similar role in the pATOM36-triggered pathway, we produced a double RNAi cell line allowing simultaneous depletion of pATOM36 and TbVCP (Fig S8). Immunoblot analyses of this cell line showed that, whereas the level of ATOM19 was slightly yet significantly stabilised upon pATOM36 and TbVCP depletion in comparison with the level found in pATOM36-depleted cells, the same was not the case for ATOM46 (Fig 3A and B). Thus, simultaneous ablation of pATOM36 and TbVCP gave essentially the same results that were observed in the pATOM36/TbMsp1 double RNAi cell line.

These results can best be explained if the depletion of one AAA-ATPase protein, TbMsp1 or TbVCP, allowed its activity to be at least partially compensated by the other. To directly test this hypothesis, we generated a triple RNAi cell line, targeting pATOM36, TbMsp1, and TbVCP simultaneously (Fig S8). With this cell line, we could show that depletion of all three proteins significantly restored the levels of ATOM19 and ATOM46 to approximately three-to-sixfold of their levels in pATOM36-depleted cells (Fig 3A, compare lanes 7 and 8). These results were independently confirmed and extended by a complementary proteomic analysis which compared the induced pATOM36 cell line (corresponding to lane 2 in Fig 3A) with the induced triple RNAi cell line depleting pATOM36, TbMsp1, and TbVCP1 simultaneously (corresponding to lane 8 in Fig 3A). In this experiment, five pATOM36 substrates and a few other OM proteins were significantly enriched more than 1.5-fold, indicating that their levels were stabilised (Fig 4, bottom panel). The simplest explanation for these results is that TbMsp1 and TbVCP have redundant, at least partially synergistic functions in the MAD pathway that lead to the degradation of pATOM36-dependent substrates.

### TbMsp1 interactors contribute to the function of the MAD pathway

Using the same approach, it was possible to test whether the four mitochondrial OM proteins that we identified to be in the same protein complex as TbMsp1 played a functional role in the MAD

A

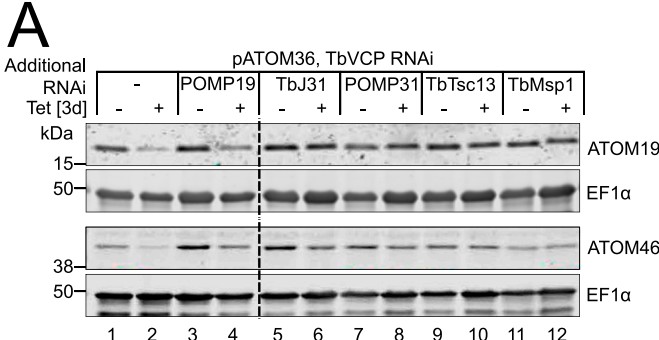

B

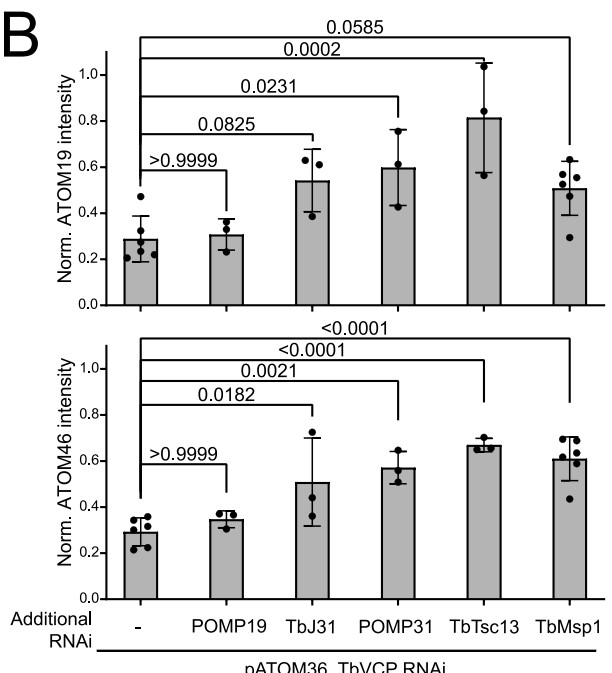

**Figure 5. TbJ31, POMP31, and TbTsc13 are required for the mitochondria-associated degradation pathway triggered by the absence of pATOM36.**
**(A)** Western blot analysis of total cellular extracts (3 × 10[6] cells each) of the indicated uninduced and induced double and triple RNAi cell lines (−/+ Tet), probed with ATOM19 and ATOM46 antisera. EF1α was used as a loading control. **(B)** Quantifications of ATOM46 and ATOM19 levels in the RNAi cell lines from immunoblots shown in Fig 3A. The signal for each sample was normalised to its respective EF1α signal and then to the respective level in uninduced cells. Data are presented as mean values with error bars corresponding to the SD (n = 3–6). The *P*-values indicated in the graph were calculated using a one-way ANOVA followed by a Bonferroni post hoc test to allow for multiple comparisons. Source data are available for this figure.

pathway investigated in this study. We constructed a series of triple RNAi cell lines, depleting either POMP31, POMP19, TbJ31, or TbTsc13 together with pATOM36 and TbVCP, to trigger the MAD pathway and to prevent pATOM36 substrates being degraded via the TbVCP-mediated arm of the pathway (Fig S8). Upon induction of RNAi, a significant restoration in the levels of ATOM19 and ATOM46 was detectable by immunoblot in triple RNAi cell lines where either TbJ31, POMP31, and TbTsc13 were depleted along with pATOM36 and TbVCP, in comparison with cells in which only pATOM36 and TbVCP were depleted (Fig 5A, compare lanes 2 with lanes 6, 8, 10, Fig 5B). In

the case of TbJ31, only the level of ATOM46 restoration was significant. This observed restoration in the levels of ATOM19 and ATOM46 upon pATOM36, TbVCP, and either TbJ31, POMP31, or TbTsc13 depletion phenocopies the effects observed in the triple RNAi cell line targeting pATOM36, TbVCP1, and TbMsp1. This strongly suggests that TbJ31, POMP31, and TbTsc13 do not only form a complex with mitochondrial TbMsp1, but that each of the three proteins also contributes to the function in the MAD pathway triggered by pATOM36 depletion. The triple RNAi cell line depleted for POMP19 did not significantly restore the levels of ATOM46 or ATOM19, suggesting that it does not affect mitochondrial TbMsp1 activity.

Under WT conditions, Msp1 forms a complex with four interacting OM proteins (Fig 1B), three of which contribute to the activity of the MAD pathway (Fig 5). We wondered if this complex formed as a response to MAD pathway activation. We performed a SILAC-pulldown experiment of the in situ HA-tagged TbMsp1 expressed in induced pATOM36-ablated RNAi cell lines. The bait Msp1 and all four Msp1-interacting OM proteins were found to be enriched to very similar extents as in WT conditions (Fig S9). Moreover, essentially the same is the case for the glycosomal proteins. Thus, the TbMsp1-containing OM protein complex described in our study is present in both the presence and absence of pATOM36.

## Discussion

We have discovered a pathway in *T. brucei* that removes destabilised α-helically anchored proteins from the mitochondrial OM. This pathway is triggered upon depletion of the OM protein biogenesis factor pATOM36 (Fig 6). Previous studies suggest that pATOM36 has two distinct functions. It mediates the integration of ATOM46 and ATOM19 into the heterooligomeric ATOM complex after the proteins have been inserted into the OM (Käser et al, 2016). However, it can also facilitate insertion of certain proteins into the OM, as was shown for POMP10 (Bruggisser et al, 2017). Removal of pATOM36 prevents integration of several ATOM subunits into the ATOM complex, leading to their degradation by the cytosolic proteasome (Fig 6). This degradation will require the selective extraction of these membrane proteins from the OM. TbMsp1 and TbVCP, the trypanosomal homologs of the Opisthokont mitochondrial quality control components Msp1 and VCP, are AAA-ATPases and therefore perfect candidates for such a job. Our results show that there is some redundancy in the system as knocking down only one of the two AAA-ATPases hardly affects the MAD pathway. Thus, this newly discovered TbMsp1 and TbVCP-linked MAD pathway likely function in safeguarding OM functions in trypanosomes, maintaining this essential interface for mitochondrial-intracellular communications.

Maintenance of protein homeostasis is essential to maintain cellular functions under both unstressed and stress conditions. Many membrane proteins in eukaryotes require selective trafficking to specific subcellular compartments and assembly into defined stoichiometric complexes before functioning. This assembly process is not 100% efficient and thus degradation of unassembled, potentially harmful complex subunits is required. Msp1 is known to extract these orphaned proteins from both the mitochondrial OM and the peroxisomal membrane, allowing their degradation by the

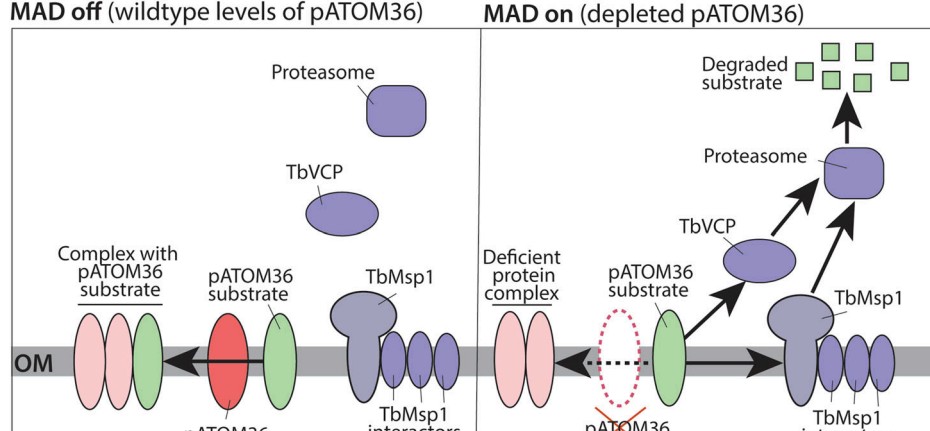

**Figure 6. Schematic model of the trypanosomal mitochondria-associated degradation (MAD) pathways triggered by the absence of pATOM36.**
Left panel, pATOM36 mediates the assembly of a subset of mitochondrial OM proteins (pATOM36 substrates) into their respective protein complexes. The mitochondrial fraction of TbMsp1 is constitutively associated with OM proteins (TbMsp1 interactors). Under these conditions, the described MAD pathways are not operational (MAD off). Right panel, RNAi-mediated ablation of pATOM36 triggers the MAD pathways (MAD on) that ultimately result in the proteasomal degradation of orphan pATOM36 substrates. This likely happens by parallel pathways linked to two different AAA-ATPases, the soluble TbVCP or the OM-integral TbMsp1. The TbMsp1-linked MAD pathway depends on three TbMsp1-interacting proteins for full activity.

proteasome (Chen et al, 2014; Hegde, 2014; Weir et al, 2017). In yeast, Msp1 has been shown to be sufficient for membrane protein extraction (Wohlever et al, 2017) or, in case of particular substrates, to function together with an interacting protein that is induced by a specific trigger, for example, in MitoCPR (Weidberg & Amon, 2018).

Trypanosomal TbMsp1 surprisingly forms a complex with at least four other integral mitochondrial OM proteins. Three of these interactors, POMP31, TbJ31, and TbTsc13, contribute to the activity of the MAD pathway that is triggered upon pATOM36 depletion.

Whereas POMP31 is only found in kinetoplastids, TbJ31 is an orthologue of the mammalian mitochondrial OM J-protein, DNAJC11, although it lacks a complete HPD motif and thus is a J-like protein (Muñoz-Gómez et al, 2015). TbJ31 and DNAJC11 both have a C-terminal DUF3395 domain suggested to mediate protein-protein interactions (Violitzi et al, 2019). It has also been reported that mammalian DNAJC11 may transiently interact with the mitochondrial contact site and cristae organizing system complex (Xie et al, 2007; Violitzi et al, 2019). TbTsc13 shows similarity to the enoyl-CoA reductase of the ER elongase complex (Miinalainen et al, 2003; Sickmann et al, 2003; Reinders et al, 2006; Parl et al, 2013). Interestingly, it has an N-terminal ubiquitin-like domain that is exposed to the cytosol (Uchida et al, 2021). As yet, we do not understand the specific role these TbMsp1-interacting proteins may play in the described MAD pathway. However, the notion that TbMsp1 may act in concert with a J-like protein that could directly or indirectly regulate chaperones seems plausible in this context. The same is the case for the ubiquitin-like domain of TbTsc13, which potentially could facilitate proteasome binding and activation (Collins & Goldberg, 2020).

How pATOM36 substrates are recognised by the MAD pathway is not yet understood; in particular, we do not know how these proteins can be recognised efficiently by both TbMsp1 and TbVCP. Msp1 substrate specificity is known to be multifaceted (Fresenius & Wohlever, 2019); however, as the pATOM36 substrates we focused on in this work, ATOM46 and ATOM19, are integral parts of the ATOM complex, we could hypothesise that these proteins become orphaned upon pATOM36 depletion, allowing them to become substrates of TbMsp1. Nevertheless, not all pATOM36 substrates are known to be components of multiprotein complexes. Cytosolic VCP is involved in diverse cellular processes, and its substrate specificity in other organisms

is governed by its numerous cofactors, many of which interact with ubiquitin conjugated to its substrates (Buchberger et al, 2015; Escobar-Henriques & Anton, 2020). The potential requirement for selective ubiquitination cascades adds another layer of yet undefined diversity to the regulation of this process.

Understanding variations in mitochondrial biogenesis across eukaryotes can provide insight into their evolution as well as into the process of how the endosymbiotic bacterial ancestor of the mitochondrion converted into an organelle. VCP and Msp1 are conserved throughout eukaryotes and, thus, were present in the last eukaryotic common ancestor (LECA). However, the convergent evolution of known divergent OM protein biogenesis factors (pATOM36, MTCH2, and MIM) for α-helically anchored OM proteins between, and even within, distinct eukaryotic supergroups suggests that LECA did not contain a protein with this function (Vitali et al, 2018).

This is in agreement with the notion that LECA contained a much simpler β-barrel-based OM protein import system (Dolezal et al, 2006; Mani et al, 2016), whereas most additional α-helical subunits of the TOM complex, for example, the receptors, were added later after a first divergence of eukaryotes to confer specificity and efficiency of the import process (Perry et al, 2006; Mani et al, 2015, 2016; Rout et al, 2021). Thus, the role of Msp1 in removing orphan α-helical OM proteins is likely not its ancestral one. Instead, the requirement of Msp1 to clear precursor blockages in the OM protein import machinery may have evolved first (Weidberg & Amon, 2018). Whether TbMsp1 has retained this activity remains to be investigated.

Thus, the TbMsp1 function linked to surveillance of OM protein biogenesis likely arose after pATOM36 evolution, and the same is the case for the mitochondrial OM protein complex formed by Msp1 and its interactors, three of which contribute to its activity. The MAD pathway triggered by the depletion of pATOM36 is, to our knowledge, the first one to be characterised in any eukaryote that is specifically linked to defects in OM protein biogenesis.

If the emergence of pATOM36 drove the evolution of a Msp1/VCP-linked pathway to survey and maintain the integrity of its activity, did the same happen in Opisthokonts? Intriguingly, there are hints that depletion of yeast MIM or mammalian MTCH2 may drive MAD pathways. Loss of these proteins does result in depletion in the level of at least some of their substrates (Vitali et al, 2018; Guna et al, 2022),

reminiscent of the proteasomal degradation of pATOM36 substrates by the MAD pathway described here. Accumulation of orphan OM proteins is likely harmful for all mitochondria, suggesting that a pathway to deal with such proteins might be required in all eukaryotes.

We therefore expect that the independent establishment of specific OM protein biogenesis pathways in different phylogenetic groups resulted in the parallel evolution of the corresponding MAD pathways in the same groups. It is likely that these systems are also connected to the widely conserved AAA-ATPases Msp1 and VCP. Should this be the case, it will be interesting to find out whether they, as with TbMsp1, also require additional factors for full activity and, if yes, what their identity might be.

There has been much progress in defining mitochondrial quality control pathways in Opisthokonts such as yeast and metazoans. However, only very recently have studies on mitochondrial quality control expanded beyond this narrow range of eukaryotic diversity. A MAD pathway has been found in trypanosomes, a member of the Discoba supergroup, that facilitates the removal of mistargeted aggregation-prone mitochondrial proteins from the cytosol (Dewar et al, 2022a). The results suggested that the depletion of cytosolic chaperones may be a general trigger of MAD throughout eukaryotes. The present Msp1 and VCP-linked pathway is the second MAD pathway discovered in trypanosomes. Further studies in other non-classical model systems are expected to improve our understanding of the fundamental features of such pathways, which are similar not because of common descent but because all eukaryotes have to cope with the shared constraints imposed by hosting mitochondria.

# Materials and Methods

**Reagents and tools table.**

| Reagent type (species) or resource | Designation | Source or reference | Identifiers | Additional information |
|---|---|---|---|---|
| Cell line (*T.brucei*) | 29.13, procyclic | Wirtz et al (1999) | | WT |
| Antibody | Anti-HA (mouse, monoclonal) | BioLegend | 901503 (MMS-101R) | WB (1:5,000), IFA (1:1,000) |
| Antibody | Anti-myc (mouse, monoclonal) | Invitrogen | 132500 | WB (1:2,000), IFA (1:50) |
| Antibody | Anti-ATOM19 | Eurogentec | Polyclonal antibody against purified protein | WB (1:500) |
| Antibody | Anti-ATOM46 | Mani et al (2015) | | WB (1:500) |
| Antibody | Anti-pATOM36 | Vitali et al (2018) | | WB (1:250) |
| Antibody | Anti-TbVCP | Gift from Prof. James Bangs, SUNY Buffalo, US | | WB (1:50) |
| Antibody | Anti-ALD | Gift from Paul Michels, University of Edinburgh, Scotland | | IFA (1:1,500) WB (1:10,000) |
| Antibody | Anti-BiP | Gift from Prof. James Bangs, SUNY Buffalo, US | | IFA (1:2,500) WB (1:50,000) |
| Antibody | Anti-TbMsp1 | Eurogentec | Peptide antibody against C + DEALKRVRPSMASSV | WB (1:1,000) |
| Antibody | Anti-ATOM40 (rabbit, polyclonal) | Niemann et al (2013) | | Bleed 1, IFA (1:1,000) |
| Antibody | Anti-VDAC (rabbit, polyclonal) | Niemann et al (2013) | | WB (1:1,000) |
| Antibody | Anti-EF1α (mouse, monoclonal) | Merk Millipore | 05-235 | WB (1:10,000) |
| Antibody | Anti-cytochrome *c* (rabbit, polyclonal) | Crausaz Esseiva et al (2004) | | WB (1:100) |
| Antibody | Anti-mitochondrial Hsp70 | Niemann et al (2013) | | WB (1:2000) |
| Antibody | Anti-ATOM69 | Mani et al (2015) | | WB (1:500) |
| Antibody | Anti-Tim9 | Niemann et al (2013) | | WB (1:100) |
| Antibody | Anti-mouse IRDye 680LT conjugated (goat) | LI-COR Biosciences | PN 926-68020 | WB (1:20,000) |

**Continued**

| Reagent type (species) or resource | Designation | Source or reference | Identifiers | Additional information |
|---|---|---|---|---|
| Antibody | Anti-rabbit IRDye 800CW conjugated (goat) | LI-COR Biosciences | PN 926-32211 | WB (1:20,000) |
| Antibody | Goat Anti-mouse Alexa Fluor 596 | Thermo Fisher Scientific | # A-11032 | IFA (1:1,000) |
| Antibody | Goat Anti-rabbit Alexa Fluor 488 | Thermo Fisher Scientific | # A-11008 | IFA (1:1,000) |
| Commercial assay or kit | Prime-a-Gene labelling kit | Promega | U1100 | Radioactive labelling of Northern probes |
| Commercial assay or kit | EZView Red Anti-c-myc affinity gel | Sigma-Aldrich | E6654 | CoIP |
| Commercial assay or kit | Anti-HA affinity matrix | Roche | 11815016001 | CoIP |
| Commercial assay or kit | Proteinase K, recombinant, PCR Grade | Roche | 3115879001 | |
| Chemical compound, drug | Tetracycline Hydrochloride | Sigma-Aldrich | T7660 | Tet |
| Chemical compound, drug | Digitonin | Biosynth | 103203 | Generation of crude mitochondrial fractions |
| Chemical compound, drug | PFA | Fluka | UN2213 | |
| Chemical compound, drug | Albumin (BSA) Fraktion V (pH 7,0) | Applichem | A1391 | |
| Chemical compound, drug | Triton X-100 | Merck Millipore | 108603 | |
| Chemical compound, drug | Tween 20 | AppliChem | A4974 | |
| Chemical compound, drug | Lysine-L U-13C, U-15N (Lys8) | Euroisotop | CNLM-291-H | SILAC labelling |
| Chemical compound, drug | Arginine-L U-13C6, U-15N4 (Arg10) | Euroisotop | CNLM-539-H | SILAC labelling |
| Chemical compound, drug | Lysine-L, 4.4.5.5-D4 (Lys4) | Euroisotop | DLM-2640 | SILAC labelling |
| Chemical compound, drug | Arginine-L 13C6 (Arg6) | Euroisotop | CLM-2265-H | SILAC labelling |
| Chemical compound, drug | Formaldeyde, light ($CH_2O$) | Sigma-Aldrich | 252549 | Peptide stable isotope dimehtyl labelling |
| Chemical compound, drug | Formaldeyde, heavy ($^{13}CD_2O$) | Sigma-Aldrich | 492620 | Peptide stable isotope dimehtyl labelling |
| Chemical compound, drug | Sodium cyano-borohydride ($NaBH_3CN$) | Sigma-Aldrich | 156159 | Peptide stable isotope dimehtyl labelling |
| Enzyme | Trypsin, MS approved | SERVA | 37286 | |
| Software, algorithm | GraphPad Prism, version 6.0 f | Graphpad software | www.graphpad.com | Depiction of growth curves and analysis of Western blot quantification |
| Software, algorithm | Fiji | ImageJ | Schindelin et al (2012) | Processing of images |
| Software, algorithm | FigureJ | ImageJ Plugin | Mutterer and Zinck (2013) | Assembly of microscopy figures |
| Software, algorithm | Image Studio Lite v. 5.2.5. | LI-COR Biosciences | | Quantification of Western blots |
| Software, algorithm | Adobe Illustrator | Adobe | www.adobe.com | Figure assembly |
| Software, algorithm | RStudio | RStudio | www.rstudio.com | Mass spectrometry analysis, volcano plotting |

## Methods and protocols

### Transgenic cell lines

Transgenic *T. brucei* cell lines were generated using the procyclic strain 29.13 (Wirtz et al, 1999). Cells were cultivated at 27°C in SDM-79 (Brun & Schonenberger, 1979) supplemented with 10% (vol/vol) FCS2, containing G418 (15 µg/ml; Gibco), hygromycin (25 µg/ml; InvivoGen), puromycin (2 µg/ml; InvivoGen), blasticidin (10 µg/ml; InvivoGen), and phleomycin (2.5 µg/ml; LifeSpan BioSciences) as required. RNAi or protein overexpression was induced by adding 1 µg/ml tetracycline to the medium.

To produce plasmids for ectopic expression of C-terminal triple c-myc- or HA-tagged TbMsp1 (Tb927.5.960), POMP31 (Tb927.6.3680), TbJ31 (Tb927.7.990), POMP19 (Tb927.10.510), and TbTsc13 (Tb927.3.1840), the complete ORFs of the respective gene were amplified by PCR and inserted in a modified pLew100 vector (Wirtz et al, 1999; Bochud-Allemann & Schneider, 2002) containing either a C-terminal triple c-myc- or HA-tag (Oberholzer et al, 2006). One TbMsp1 allele was tagged in situ at the C-terminus with a triple HA-tag via a PCR approach, using a pMOTag vector containing a phleomycin resistance cassette as described in Oberholzer et al (2006).

RNAi cell lines were prepared using a pLew100-derived vector with a 500 bp target gene fragment and its reverse complement present with a 460 bp stuffer in-between, generating a stem-loop construct. The RNAis targeted the indicated nucleotides (nt) of the ORF of proteasome subunit $\beta_1$ (nt 266–759), TbVCP (nt 423–896), TbTsc13 (nt 379–891), TbJ31 (nt 831–1,255), POMP31 (nt 170–568), POMP19 (nt 182–610), TbMsp1 (nt 530–940). The pATOM36 RNAi construct was previously published (Pusnik et al, 2012).

### Digitonin extraction

Cell lines were induced for 16 h before the experiment to express the epitope-tagged proteins. Crude mitochondria-enriched fractions were obtained by incubating $1 \times 10^8$ cells on ice in 0.6 M sorbitol, 20 mM Tris–HCl (pH 7.5), and 2 mM EDTA (pH 8) containing 0.015% (wt/vol) digitonin for the selective solubilization of plasma membranes. Centrifugation (5 min, 6,800 g, 4°C) yielded a cytosolic supernatant and a mitochondria-enriched pellet. Equivalents of $1.3 \times 10^6$ cells of each fraction were analysed by SDS–PAGE and subsequent Western blotting to demonstrate organellar enrichment for proteins of interest.

### Alkaline carbonate extraction

To separate soluble or peripherally membrane-associated proteins from integral membrane proteins, a mitochondria-enriched pellet was generated as described above by digitonin extraction and resuspended in 100 mM $Na_2CO_3$ (pH 11.5). Centrifugation (10 min, 100,000g, 4°C) yielded a supernatant containing soluble proteins and a pellet containing membrane fragments. Equivalents of $7.5 \times 10^6$ cells of each fraction were subjected to SDS–PAGE and immunoblotting.

### Proteinase K protection assay

A mitochondria-enriched digitonin pellet from $5 \times 10^7$ cells over-expressing C-terminally tagged Msp1, POMP31, TbJ31, or TbTsc13 was generated as described above. The pellet was resuspended in 250 mM sucrose, 80 mM KCl, 5 mM MgAc, 2 mM KH2PO4, and 50 mM Hepes, and distributed in five equal samples. Triton X-100 was added to indicated samples to 0.5% (vol/vol). Proteinase K was added to the samples in concentrations as indicated. After 15 min incubation on ice, the reactions were stopped by adding PMSF to 5 mM. Samples without Triton X-100 were centrifuged (3 min, 6,800 g, 4°C) and all samples were resuspended in SDS loading buffer. In each sample, $1 \times 10^6$ cell equivalents were subjected to SDS–PAGE and Western blotting.

## Immunoprecipitation

Digitonin-extracted mitochondria-enriched fractions of $1 \times 10^8$ induced cells were solubilized on ice in 20 mM Tris–HCl (pH 7.4), 0.1 mM EDTA, 100 mM NaCl, 25 mM KCl, 1x protease inhibitor mix (EDTA-free; Roche), and 1% (wt/vol) digitonin. After centrifugation (15 min, 20817g, 4°C), the lysate (IN, input) was transferred to either 50 µl of HA bead slurry (anti-HA affinity matrix; Roche) or 50 µl c-myc bead slurry (EZview red anti-c-myc affinity gel; Sigma-Aldrich), both of which had been equilibrated in wash buffer (20 mM Tris–HCl [pH 7.4], 0.1 mM EDTA, 100 mM NaCl, 10% glycerol, 0.2% [wt/vol] digitonin). After incubating at 4°C for 2 h on a rotating wheel, the supernatant containing the unbound proteins (FT, flow through) was removed. The bead slurry was washed three times with wash buffer. Bound proteins were eluted by boiling the resin in 60 mM Tris–HCl (pH 6.8) containing 2% SDS (IP). 2.5% of crude mitochondrial fractions (Input, IN), unbound proteins in the flow through (FT), and 50% of the final eluates (IP) were separated by SDS–PAGE and analysed by Western blot.

### SILAC immunoprecipitations

Cells were grown for 5 d in SILAC medium (SDM80 containing 5.55 mM glucose, supplemented with 10% dialyzed, heat-inactivated FCS, 7.5 mg/l hemin) containing isotopically distinct variants of arginine ($^{12}C_6^{14}N_4$/Arg0, $^{13}C_6^{14}N_4$/Arg6, or $^{13}C_6^{15}N_4$/Arg10; 226 mg/l each) and lysine ($^{12}C_6^{14}N_2$/Lys0, $^{12}C_6^{14}N_2^2H_4$/Lys4, or $^{13}C_6^{15}N_4$/Lys8; 73 mg/l each) (Eurisotope). $2 \times 10^8$ WT cells and cells expressing in situ tagged Msp1-HA (in the presence or absence of pATOM36) were mixed and washed with 1x PBS. Crude mitochondria-enriched fractions were obtained by digitonin extraction as described above. The pellet of the digitonin extraction was subjected to immunoprecipitation as described above. Proteins were precipitated after the methanol-chloroform protocol (Wessel & Flügge, 1984) and further processed for liquid chromatography-mass spectrometry (LC-MS) analysis including reduction in cysteine residues, alkylation of thiol groups, and tryptic digestion as described before (Dewar et al, 2022b). The experiment was performed in three biological replicates with different labelling schemes.

### RNA extraction and Northern blotting

Acid guanidinium thiocyanate-phenol-chloroform extraction according to Chomczynski and Sacchi (1987) was used for isolation of total RNA from uninduced and induced RNAi cells. Total cellular RNA was separated on a 1% agarose gel in 20 mM MOPS buffer supplemented with 0.5% formaldehyde. Northern probes were generated from gel-purified PCR products corresponding to the RNAi inserts and radioactively labelled using the Prime-a-Gene labelling system (Promega).

### Immunofluorescence microscopy

Induced ×10[6] cells overexpressing the indicated tagged proteins were harvested by centrifugation (5 min, 1,800*g*) and washed with 1x PBS. After resuspension in 1x PBS, the cells were left adhering on a glass slide in a wet chamber. The cells were fixed with 4% PFA, permeabilised with 0.2% Triton X-100, and blocked with 2% BSA in 1x PBS. Antibodies were incubated on the slides in 1% BSA and 1x PBS. The dried slides were mounted with Vectashield containing 4 DAPI (Vector Laboratories, P/N H-1200). Images were acquired with a DFC360 FX monochrome camera (Leica Microsystems) mounted on a DMI6000B microscope (Leica Microsystems). Image analysis and deconvolution were performed using LASX software (version 3.6.20104.0; Leica Microsystems). The acquired images were processed using Fiji (ImageJ version 2.10./1.53; Java 1.8.0_172 [64 bit]). The Pearson product-moment correlation coefficient (Pearson's *r*) was calculated for a region of interest defined as one representative cell that is shown in the Figure using Fiji's Coloc 2 analysis. Microscopy figures were composed using FigureJ (Mutterer & Zinck, 2013).

### Peptide stable isotope dimethyl labelling and high-pH reversed-phase fractionation

RNAi cell lines were grown in triplicate in SDM-79 for 3 d, in the presence or absence of tetracycline. 1 × 10[8] cells were centrifuged (8 min, 1,258*g*, RT) and washed with 1x PBS. The pellets were flash frozen in liquid nitrogen and subsequently processed for tryptic in-solution digestion as described before (Peikert et al, 2017). Dried peptides were reconstituted in 100 mM tetraethylammonium bicarbonate, followed by differential labelling with "light" or "heavy" formaldehyde ($CH_2O$/$^{13}CD_2O$; Sigma-Aldrich) and sodium cyanoborohydride ($NaBH_3CN$; Sigma-Aldrich) (Morgenstern et al, 2021). Labelling efficiencies (>99% for all individual experiments) were determined by LC-MS analysis. Equal amounts of differentially "light" and "heavy" labelled peptides derived from the respective control and induced RNAi cells were mixed, purified, and fractionated by high pH reversed-phase chromatography using StageTips essentially as described previously (von Känel et al, 2020). In brief, peptides, reconstituted in 10 mM $NH_4OH$, were loaded onto StageTips and eluted stepwise with 0%, 2.7%, 5.4%, 9.0%, 11.7%, 14.4%, 36%, and 65% (vol/vol each) acetonitrile (ACN)/10 mM $NH_4OH$. Fractions 1 and 7 (0% and 36% ACN eluates) and fractions 2 and 8 (2.7% and 65% ACN eluates) were combined for LC-MS analysis.

### Quantitative LC-MS analysis

Before LC-MS analysis, peptides were desalted using StateTips, vacuum-dried, and reconstituted in 0.1% (vol/vol) trifluoroacetic acid. LC-MS analyses were performed using either a Q Exactive Plus (Msp1-HA SILAC IPs) or an Orbitrap Elite (RNAi experiments) mass spectrometer connected to an UltiMate 3,000 RSLCnano HPLC system (all instruments from Thermo Fisher Scientific). Peptides were loaded and concentrated on PepMap C18 precolumns (length, 5 mm; inner diameter, 0.3 mm; Thermo Fisher Scientific) at a flow rate of 30 *µ*l/min and separated using Acclaim PepMap C18 reversed-phase nano-LC columns (length, 500 mm; inner diameter, 75 *µ*m; particle size, 2 *µ*m; pore size, 100 Å; Thermo Fisher Scientific) at a flow rate of 0.25 *µ*l/min. The solvent system used for the elution of peptides from Msp1-HA SILAC IP experiments consisted of 0.1%

(vol/vol) formic acid (FA; solvent A1) and 86% (vol/vol) ACN/0.1% (vol/vol) FA (solvent B1). The following gradient was applied: 4–39% solvent B1 in 195 min followed by 39–54% B1 in 15 min, 54–95% B1 in 3 min, and 5 min at 95% B1. For the elution of peptides from RNAi experiments, 4% (vol/vol) dimethyl sulfoxide (DMSO)/0.1% (vol/vol) FA (solvent A2) and 48% (vol/vol) methanol/30% (vol/vol) ACN/4% (vol/vol) DMSO/0.1% (vol/vol) FA (solvent B2) were used. A gradient ranging from 3–65% solvent B2 in 65 min, 65–80% B2 in 5 min, and 5 min at 80% B2 was applied.

Mass spectrometric data were acquired in a data-dependent mode. The Q Exactive Plus was operated with the following settings: mass range, *m/z* 375 to 1,700; resolution, 70,000 (at *m/z* 200); target value, 3 × 10[6]; and maximum injection time (max. IT), 60 ms for MS survey scans. Fragmentation of up to 12 of the most intense multiply charged precursor ions by higher energy collisional dissociation was performed with a normalised collision energy (NCE) of 28%, a target value of 10[5], a max. IT of 120 ms, and a dynamic exclusion (DE) time of 45 s. The parameters for MS analyses at the Orbitrap Elite were as follows: mass range, *m/z* 370 to 1,700; resolution, 120,000 (at *m/z* 400); target value, 10[6]; and max. IT, 200 ms for survey scans. A TOP15 (pATOM36/subunit $β_1$ double and pATOM36/TbVCP/TbMsp1 triple RNAi experiments) or TOP25 (pATOM36 RNAi experiments) method was applied for fragmentation of multiply charged precursor ions by low energy collision-induced dissociation in the linear ion trap (NCE, 35%; activation q, 0.25; activation time, 10 ms; target value, 5,000; max. IT, 150 ms; DE, 45 s).

Proteins were identified and quantified using MaxQuant/Andromeda (Cox & Mann, 2008; Cox et al, 2011) (version 1.5.5.1 for Msp1-HA SILAC IP and 1.6.0.1 for RNAi data). Mass spectrometric raw data were searched against a TriTryp database specific for *T. brucei* TREU927 (release version 8.1 for Msp1-HA SILAC IP and 36 for RNAi data; downloaded from https://tritrypdb.org). For protein identification, MaxQuant default settings were applied, with the exception that only one unique peptide was required. For relative quantification, the appropriate settings for SILAC labelling (light labels, Lys0/Arg0; medium-heavy, Arg6/Lys4; heavy, Lys8/Arg10) or stable isotope dimethyl labelling (light, dimethylLys0/dimethylNterLys0; heavy, dimethylLys6/dimethylNterLys6) were chosen. Quantification was based on at least one ratio count. The options "match between runs" and "requantify" were enabled. Only proteins quantified in at least two independent replicates per dataset were considered for further analysis. The mean $\log_{10}$ (SILAC IP data) or mean $\log_2$ (RNAi data) of protein abundance ratios was determined, and a one-sided (SILAC IP data) or two-sided (RNAi data) *t* test was performed. For information about the proteins identified and quantified, see Table S1 (TbMsp1-HA SILAC IPs) and Table S2 (RNAi experiments) in the PRIDE database.

### Computational analysis of proteins

Conserved structural elements of Msp1 (Ogura et al, 2004; Martin et al, 2008; Wang et al, 2020) are highlighted in Fig 1A. TMDs were predicted using Phobius (Madeira et al, 2022) (TbMsp, POMP31, TbJ31, POMP19) or HMMTOP (Tusnády & Simon, 1998) (TbTsc13), and conserved domains were either predicted with ncbi.nlm.nih.gov/Structure (POMP19, TbTsc13) or annotated Pfam domains on HMMER (Potter et al, 2018) (TbMsp1, TbJ31). The ubiquitin-like domain of TbTsc13 was predicted by HHpred (Zimmermann et al, 2018). The multiple amino acid sequence alignment of TbMsp1, ATAD1 from

*H. sapiens* (HsATAD1), and Msp1 from *S. cerevisiae* (ScMsp1) shown in Fig S1 was performed with Clustal Omega (Sievers et al, 2011).

## Data Availability

The mass spectrometry data have been deposited to the ProteomeXchange Consortium (Deutsch et al, 2020) via the PRIDE (Perez-Riverol et al, 2022) partner repository and are accessible using the dataset identifiers PXD039631 (SILAC IP data) and PXD039634 (RNAi data).

## Supplementary Information

## Acknowledgements

We thank Bettina Knapp for technical assistance. We thank Noemis Zbären and Gabriel Klesse for help at the initial stage of the project. Work in the laboratory of B Warscheid was supported by the Deutsche Forschungsgemeinschaft (DFG, German Research Foundation) project ID 403222702/SFB 1381 and Germany's Excellence Strategy (CIBSS—EXC-2189—Project ID 390939984). Work in the laboratory of A Schneider was supported in part by NCCR RNA and Disease, a National Centre of Competence in Research (grant number 205601), and by project grant SNF 205200, both funded by the Swiss National Science Foundation.

### Author Contributions

M Gerber: conceptualization, investigation, methodology, and writing—original draft, review, and editing.
I Suppanz: formal analysis and investigation.
S Oeljeklaus: formal analysis and investigation.
M Niemann: investigation and writing—review and editing.
S Käser: investigation.
B Warscheid: conceptualization, supervision, funding acquisition, and writing—review and editing.
A Schneider: conceptualization, supervision, funding acquisition, and writing—original draft, review, and editing.
CE Dewar: conceptualization, supervision, investigation, methodology, and writing—original draft, review, and editing.

### Conflict of Interest Statement

The authors declare that they have no conflict of interest.

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
