## [Reviewer comments · Life Science Alliance]

Life Science Alliance

A Msp1-containing complex removes orphaned proteins in the mitochondrial outer membrane of *T. brucei*

Markus Gerber, Ida Suppanz, Silke Oeljeklaus, Moritz Niemann, Sandro Käser, Bettina Warscheid, André Schneider, and Caroline Dewar

DOI: <https://doi.org/10.26508/lsa.202302004>

Corresponding author(s): André Schneider, University of Bern; Bettina Warscheid, Faculty of Chemistry and Pharmacy, Biochemistry II, Theodor Boveri-Institute, University of Würzburg; and Caroline Dewar, University of Bern

Review Timeline:

Submission Date:	2023-02-22
Editorial Decision:	2023-03-27
Revision Received:	2023-06-30
Editorial Decision:	2023-07-25
Revision Received:	2023-07-27
Accepted:	2023-07-31

Scientific Editor: Novella Guidi

Transaction Report:

March 27, 2023

Re: Life Science Alliance manuscript #LSA-2023-02004-T

Prof. André Schneider
University of Bern
Department of Chemistry, Biochemistry and Pharmaceutical Science
Freiestr. 3
Bern 3012
Switzerland

Dear Dr. Schneider,

Thank you for submitting your manuscript entitled "A Msp1-containing complex removes orphaned proteins in the mitochondrial outer membrane of trypanosomes" to Life Science Alliance. The manuscript was assessed by expert reviewers, whose comments are appended to this letter. We invite you to submit a revised manuscript addressing the Reviewer comments.

Thank you for this interesting contribution to Life Science Alliance. We are looking forward to receiving your revised manuscript.

Sincerely,

B. MANUSCRIPT ORGANIZATION AND FORMATTING:

Reviewer #1 (Comments to the Authors (Required)):

The authors argue about the role of TbMsp1 in quality control of mitochondrial OM proteins along with TbVCP and 3 other proteins which varies from its homolog in yeast and mammals.

Major points:

1. In Fig. 2A isotype controls should be included. Panel1/2nd row the TbMsp1 crop should be redone for the POMP19-TbMsp1 blot.
2. For validation of fractionation purity of the P1 fraction a glycosomal and ER marker should be included in Fig. S2A. it would be nice if the fractionation of certain proteins which are known to be exclusively glycosomal are shown using method. Moreover, it might be more informative to include proteinase K digestion and it is performed on P1 samples with some OM and mitochondrial matrix specific markers to validate the intactness of the mitochondria.
3. For all the fluorescence experiments quantification of co-localization (like Mander's coefficient) could be performed and control cells without any protein expression should be included to show the null morphology of structures under investigation. In addition, Fig S2B should have glycosomal and ER marker staining. Fig S3 should show mitochondrial staining using particular antibodies, DAPI staining should not be used to depict mitochondria because it can be artifactual.

Minor points:

1. Fig 3A and 4A, given the variation in ATOM19 and ATOM46 between these 2 blots for same conditions (compare pATOM36+TbVCP RNAi) it would be more uniform if WT samples can be included for these blots.
2. Fig S4, S5, error bars are either absent or not visible.

Reviewer #2 (Comments to the Authors (Required)):

Summary

Gerber et al. characterize *T. brucei* Msp1, arguing that the study of mitochondrial quality control from the *Discoba* supergroup will provide evolutionary insights that complement work with Opisthokonts eukaryotic supergroup, which includes *S. cerevisiae* and *H. sapiens*. Using a combination of mass spectrometry and immunoprecipitation experiments, the authors identified several interaction partners for Msp1 and chose to focus on four outer mitochondrial membrane proteins (POMP31, TbJ31, TbTsc13, and POMP19). The authors do a good job validating that these are indeed outer membrane proteins, with some glycosome localization Tsc13. The authors also use inducible RNAi experiments to show that Msp1 and VCP are involved in promoting the degradation of orphaned ATOM19 and ATOM46, likely through parallel pathways. The authors then bring the story full circle by showing that knockdown of TbJ31, Pomp31 and Tsc13 can phenocopy Msp1 knockdown. I see three major conclusions in the paper: 1) Msp1 forms a stable complex with POMP31, TbJ31, TbTsc13, and POMP19, 2) the aforementioned proteins are required for full Msp1 activity, and 3) Msp1 and VCP operate in parallel pathways to remove orphaned ATOM subunits after pATOM36 knockdown. While the insights are interesting, the paper suffers from overinterpretation of poorly controlled experiments and there is insufficient evidence to support conclusions 1 & 2. I believe the paper has merit and can be quickly published in a more descriptive format that significantly scales back the claims and acknowledges experimental limitations. Alternatively, I have described control experiments that are necessary to support the conclusions of the paper as it is currently written.

Major Concerns

While the results clearly show that POMP31, TbJ31, TbTsc13, POMP19, and Msp1 interact, I am not convinced that they form a stable complex. It is critical to differentiate between Msp1 substrates and stable interaction partners. The argument for a stable complex hinges on the pulldowns in Figure 2. While many pulldowns are reciprocal, not all of them are. There are several technical issues with these pulldowns, including the use of overexpression vectors, non-specific SDS PAGE elution, and inadequate negative controls, which could explain the lack of fully reciprocal results. A few more control experiments are necessary to solidify these results. In particular, the authors should perform negative control experiments with a strain expressing only one of the tagged proteins to test for non-specific interaction with the resin. For example, a strain with only Msp1-HA should be subjected to an anti-myc pulldown. I appreciate that VDAC could potentially serve as a negative control, but a beta-barrel protein like VDAC could have significantly different propensities for non-specific interaction with resin than the

alpha-helical proteins of interest. Another concern with the pull-downs is that overexpression of Msp1 in *S. cerevisiae* is toxic. Is a similar phenotype observed in *T. brucei*?

Another way to validate a stable complex is to use RNAi knockdown. For many stable complexes, the knockdown of one protein can destabilize the whole complex. Does this also occur with the proposed TbMsp1-POMP19-POMP31-TbJ31-TbTsc13 complex? As the inducible RNAi strains are already made (Figure S4), many of the tools required for this experiment are readily available.

Since knockdown of pATOM36 affects assembly of ATOM, this can lead to pleiotropic phenotypes and/or indirect effects that will complicate data interpretation. Similarly, knockdown of VCP can also lead to pleiotropic phenotypes. With the pATOM36/VCP double knockdown many factors are changed in the cell and I am very uneasy drawing mechanistic models from simple phenocopy experiments. For example, the authors state "POMP31, TbJ31 and TbTsc13 contribute to TbMsp1 activity in the MAD pathway that is triggered upon pATOM36 depletion", yet the data in Figure 4 lacks sufficient controls to support this statement. This conclusion depends on knockdown of TbJ31, POMP31, TbTsc13 phenocopying Msp1 knockdown in the pATOM36/VCP double knockdown background. I believe that these knockdowns phenocopy Msp1 knockdown, but there are insufficient controls to conclude that they phenocopy is the result of disrupting Msp1 activity. For example, there are no controls to show that this phenotype depends on VCP knockdown, as the model would predict. Likewise, the model predicts that additional knockdown of Msp1 should have no additional effect on ATOM19/46 levels. Unfortunately, neither of these controls were performed.

In the same vein, the figure title for Figure 4 states "TbJ31, POMP31, and TbTsc13 are required for the degradation of pATOM36 substrates by TbMsp1". The authors do not demonstrate that these proteins are required for Msp1 activity; they only show that knockdown of these proteins phenocopies Msp1 knockdowns. As described above, given the pleiotropic effects of pATOM36 and VCP knockdown, there are many possible explanations for this observation.

In the discussion section the authors state "This degradation requires their selective extraction from the OM by the trypanosomal homologs of the Opisthokont mitochondrial quality control components Msp1 or VCP." The authors have not shown extraction, they have only shown Msp1/VCP dependent degradation in whole cell extracts, not from mitochondria. To show extraction the authors would need to show a change in mitochondrial concentration (integral, not peripheral) that is Msp1/VCP dependent and also show direct interaction of ATOM19 and ATOM46 with Msp1. Demonstrating decreased interaction of Msp1 with ATOM19/46 after knockdown of TbJ31, POMP31, and TbTsc13 could also support the model that these proteins are required for Msp1 activity.

TbJ31 expression is significantly decreased upon pATOM36 knockdown (Figure 3C). But further knockdown of TbJ31 rescues the effects of pATOM36 knockdown (Figure 4). Since the authors already have the mass spec data in Figure 3C, they should explicitly say how knockdown of pATOM36 affects the expression levels of POMP31, TbJ31, TbTsc13, POMP19 and Msp1 in the text and in Figure 3C.

Minor Concerns

- In Figure 2 with the POMP19-HA and TbMsp1-myc pull down, it appears that there is a non-specific band in the Myc blot. I'm not bothered by the non-specific band, but rather that the crop occurs in the middle of the band. The authors should reformat the figure and specifically label the band as non-specific to facilitate easy data interpretation.
- In Figure S4, why is there is size shift in the EtBr stained rRNAs for POMP19?
- In Figure S5, with the TbMsp1/pATOM36 and TbJ31/pATOM36/TbVCP knockdowns, there appears to be a non-specific band in the pATOM36 blots, but this isn't present in the other blots. Can the authors explain this? I think there are arrows to the bands of interest, but it is very difficult to see this. Bands of interest and non-specific bands should both be clearly labeled.
- In Figure S5, there appears to be a non-specific band in the TbMsp1 blot that is not present in Figure S4.

Reviewer #3 (Comments to the Authors (Required)):

Sustaining normal mitochondrial functions hinges on the presence of sophisticated mitochondrial-associated degradation (MAD) pathways that survey the OMM and monitor its protein precursors for proper targeting and assembly. The highly conserved Msp1, an ATPase associated with diverse cellular activities, plays an instrumental role in this process. In this manuscript the authors report the findings that TbMsp1, the *Trypanosoma brucei* ortholog of Msp1, is localized to both the mitochondrial OM and glycosomes, a kinetoplastid-specific organelle related to peroxisomes. Further, the authors demonstrated that cells depleted of pATOM36, which mediates complex assembly of α -helically anchored mitochondrial outer membrane proteins, results in TbMsp1 mediating the removal of destabilized OM proteins in concert with cytosolic TbVCP by targeting them to the proteasome. The authors report the identification of four integral OM proteins that stably interact with TbMsp1, and show that, uniquely, three of these interaction partners are required for the full activity of TbMsp1 in this quality control pathway.

Overall, this is an interesting study, and the authors indeed reported a number of important observations. While overall, I would

be supportive of publication, there are some points that need be addressed prior to further consideration.

1- The authors show, in their subcellular fractionation of cells expressing TbMsp1-HA and its four epitope-tagged OM interactors, that they all co-fractionate with the voltage dependent anion channel (VDAC) (Fig S2A), as would be expected for mitochondrial OM proteins. It would be informative if a proteinase K-treatment assay (in addition to the carbonate extraction assay shown) is performed on the mitochondrial solubilized fractions to faithfully confirm that all the 4 interacting proteins are predominantly integral membrane proteins.

2- The authors have demonstrated in figure 1 that TbMsp1 interacts with proteins of the mitochondrial OM and the glycosomes. What is the trigger for this complex formation? Is it constitutive or inducible? Have the authors tried to investigate and monitor the complex formation under the condition of pATOM36 depletion with respect to WT condition? Is the formation of this complex reversible or irreversible? It would be interesting to better clarify these points to faithfully shed the light on the dynamics of this complex formation and to appreciate the context of its potential biological role.

3- In figure 1B, the authors are advised to increase the figure size and resolution to make the figure a bit clearer for the reader.

4- In Figure 3, the authors (correctly, I believe) focus on changes in the levels of 2 of 11 OM proteins that are reduced in pATOM36 RNAi. It would be good to have some discussion about the identity and putative functions of the 9 other proteins. Subsequently, they show that several pATOM36 substrates are increased/rescued by RNAi against the proteasome or combined RNAi against MSP1 and VCP. However, in the volcano plots in Figure 3C, there are many additional proteins that appear significantly changed with pATOM36 RNAi (top) or with combined proteasome (middle) or MSP1/VCP (bottom) RNAi. Some analysis (i.e. string or pathway analysis, etc...) and discussion about their function should be provided.

5- A model figure to recapitulate their findings would be useful.

6- There are a few minor typos that needs attention from the authors. For instance, in the introduction, second paragraph, line 4 (plays a key role in in this process). "in" is duplicated.

Reviewer #4 (Comments to the Authors (Required)):

Gerber et al. presented interesting findings about the role of Msp1- containing complex in mitochondrial outer membrane protein quality control. They found that:

- TbMsp1 interacts with four MOM proteins possibly forming a complex
- TbMsp1 involves in degradation of MOM proteins via proteasomal degradation
- TbMsp1 and its interactors involves in degradation of MOM proteins via MAD pathway

While the referee finds that the findings presented in this study are compelling and potentially would be important to advance knowledge in mitochondrial proteostasis in *T. brucei*, the experimental set ups used in the study are not optimal (see comments below) and might compromise the findings.

Here are important concerns:

1. The western blot interaction analyses shown in Figure 2A are missing an important control. In the current set up it is impossible to judge whether most of the faint bands detected in the IP fractions were background or interactions. VDAC cannot be used as a control here since different antibodies were used, especially because the different preys were overexpressed. Ideally something like an Ig control should be used here to conclude the dataset.
2. Cellular protein levels are driven by their synthesis and degradation. To monitor the effect of degradation on protein levels, treatment to shut down protein synthesis, such as with Cycloheximide, is important. Otherwise changes in protein levels can be interpreted as changes in either synthesis or degradation or both. Thus, throughout the manuscript (Fig 3 & 4) such treatment need to be incorporated to conclude any effect on protein degradation. Otherwise the referee is not convinced by the conclusion presented in this study.
3. The authors offer a model, where TbMsp1 and their interactors are involved in mediating protein extraction from MOM and lead to their degradation by proteasome in the cytosol. To state this, similar experiments to Fig 3 and 4 need to be done in purified mitochondria to see whether depletion of TbMsp1 complex affects extraction of MOM proteins for degradation. Otherwise, the authors need to tone down the speculation that the substrates proteins are actually need to be extracted from MOM.

The referee feels that these concerns need to be addressed first for this study to be considered as a publication in Life Science Alliance journal.

Reviewer comments are in italics, Our responses are underlined

Reviewer #1 (Comments to the Authors (Required)):

The authors argue about the role of TbMsp1 in quality control of mitochondrial OM proteins along with TbVCP and 3 other proteins which varies from its homolog in yeast and mammals.

Major points:

1. In Fig. 2A isotype controls should be included. Panel 1/2nd row the TbMsp1 crop should be redone for the POMP19-TbMsp1 blot.

We do not understand what is meant by isotype controls. However, we added an additional control suggested by reviewer 2 which may help to settle this issue. Cell lines individually expressing TbTsc13-HA, POMP19-HA, POMP31-myc, TbJ31-myc, TbMsp1-myc were subjected to pulldown with anti HA- and myc-beads, respectively. The results show that, as expected, the tagged proteins were only recovered in the pellet when using resin with matching anti-HA or myc beads, respectively. Thus, no unspecific interaction with the resins was observed (see new Fig. S6A).

The new control experiments are now discussed in the revised manuscript (highlighted, page 5 and 6).

The cropping problem with Panel 1/2nd row was caused by the probing and reprobing of the same immunoblot with two different antibodies detecting proteins of very similar molecular weight. The experiment has now been repeated with duplicate immunoblots and the panel in question has been replaced (see revised Fig. 2).

2. For validation of fractionation purity of the P1 fraction a glycosomal and ER marker should be included in Fig. S2A. it would be nice if the fractionation of certain proteins which are known to be exclusively glycosomal are shown using method. Moreover, it might be more informative to include proteinase K digestion and it is performed on P1 samples with some OM and mitochondrial matrix specific markers to validate the intactness of the mitochondria.

We probed a digitonin extraction shown with the ER marker BiP and the glycosomal marker aldolase. The results in revised Fig. S2B shows that BiP preferentially fractionates with the pellet. Aldolase was found solely in the pellet fraction, similar to the mitochondrial marker VDAC.

We did the proteinase K protection assay with cells expressing C terminally tagged Msp1, POMP31, TbJ31 and TbTsc13 and got the expected results. Msp1 and the three interacting proteins which influence the activity of the MAD pathway are sensitive to the proteinase K digestion similar to the OM marker ATOM69, whereas the intermembrane space localized Tim9 and the matrix maker mHsp70 are only digested after the membranes have been dissolved by Triton (revised Fig. S2C).

The manuscript has been modified accordingly. A new paragraph describing the findings has been added (highlighted, page 5).

3. For all the fluorescence experiments quantification of co-localization (like Mander's coefficient) could be performed and control cells without any protein expression should be included to show the null morphology of structures under investigation.

We repeated the entire immunofluorescence analysis (IFA) adding a postprocessing deconvolution step. This resulted in higher quality images showing clearer localization of the tested antigens. Moreover, we have now quantified the extent of co-localization of all four interactors with mitochondrial ATOM40, glycosomal aldolase and with the ER marker BiP using the Pearson's R value. The results are in agreement with the reported localizations: TbMsp1 (Mito/Glyco), POMP31 (Mito), Tbj31 (Mito), POMP19 (Mito), TbTsc13 (Mito/ER). Finally, we have added IFA pictures stained with ATOM40, aldolase and BiP antisera of the corresponding parent cell line that does not express any tagged protein (see revised Fig. S3ABC).

We would like to emphasize that the localization of all four proteins in question have been analyzed by three independent methods all of which gave consistent results:
1)Crude fractionation using low concentration of digitonin (which indicates whether a protein is in the cytosol or in the particulate fraction).
2)Quantification of IFA which can be used to distinguish between mitochondria, glycosomes and the ER.
3)Previously published proteomic profiles which can be used to distinguish between mitochondrial OM and ER localization.

In addition, Fig S2B should have glycosomal and ER marker staining.

This has now been done. The localization of all interactors in the IFA has been compared to mitochondrial, glycosomal and ER Markers (see revised Fig. S3ABC).

Fig S3 should show mitochondrial staining using particular antibodies, DAPI staining should not be used to depict mitochondria because it can be artifactual.

See comment above. – For technical reasons, we can only analyze two antibodies simultaneously (in addition to DAPI). – The DAPI-stain was not intended to be marker for mitochondria.

Minor points:

1. Fig 3A and 4A, given the variation in ATOM19 and ATOM46 between these 2 blots for same conditions (compare pATOM36+TbVCP RNAi) it would be more uniform if WT samples can be included for these blots.

Fig. 3 specifically focuses on the role of TbVCP and TbMsp1 in the MAD pathway, which is why we used the pATOM36 RNAi cell line as a control. In Fig. 4, we intended to selectively measure the effect that additional ablation of the four Msp1 interacting proteins has on the MAD pathway. We believe that comparing it with the pATOM36/TbVCP RNAi double cell line is the fairest control we can do. In both figures, to quantify, we normalised the level of substrates in each of the RNAi cell lines analysed induced vs non induced to ensure differences in basal expression levels were accounted for, and growth curves shown in Fig. S8 show the RNAi induction in these cell lines is not leaky.

2. Fig S4, S5, error bars are either absent or not visible.

Cells numbers can be measured very accurately using a Coulter Counter. This results in error bars that are too small to be displayed. This is now described in the legend of the corresponding Figures (see Fig. S7, S8).

Reviewer #2 (Comments to the Authors (Required)):

Summary

Gerber et al. characterize *T. brucei* Msp1, arguing that the study of mitochondrial quality control from the *Discoba* supergroup will provide evolutionary insights that complement work with *Opisthokonts* eukaryotic supergroup, which includes *S. cerevisiae* and *H. sapiens*. Using a combination of mass spectrometry and immunoprecipitation experiments, the authors identified several interaction partners for Msp1 and chose to focus on four outer mitochondrial membrane proteins (POMP31, TbJ31, TbTsc13, and POMP19). The authors do a good job validating that these are indeed outer membrane proteins, with some glycosome localization Tsc13. The authors also use inducible RNAi experiments to show that Msp1 and VCP are involved in promoting the degradation of orphaned ATOM19 and ATOM46, likely through parallel pathways. The authors then bring the story full circle by showing that knockdown of TbJ31, Pomp31 and Tsc13 can phenocopy Msp1 knockdown. I see three major conclusions in the paper: 1) Msp1 forms a stable complex with POMP31, TbJ31, TbTsc13, and POMP19, 2) the aforementioned proteins are required for full Msp1 activity, and 3) Msp1 and VCP operate in parallel pathways to remove orphaned ATOM subunits after pATOM36 knockdown. While the insights are interesting, the paper suffers from overinterpretation of poorly controlled experiments and there is insufficient evidence to support conclusions 1 & 2. I believe the paper has merit and can be quickly published in a more descriptive format that significantly scales back the claims and acknowledges experimental limitations. Alternatively, I have described control experiments that are necessary to support the conclusions of the paper as it is currently written.

Major Concerns

While the results clearly show that POMP31, TbJ31, TbTsc13, POMP19, and Msp1 interact, I am not convinced that they form a stable complex. It is critical to differentiate between Msp1 substrates and stable interaction partners. The argument for a stable complex hinges on the pulldowns in Figure 2. While many pulldowns are reciprocal, not all of them are. There are several technical issues with these pulldowns, including the use of overexpression vectors, non-specific SDS PAGE elution, and inadequate negative controls, which could explain the lack of fully reciprocal results. A few more control experiments are necessary to solidify these results. In particular, the authors should perform negative control experiments with a strain expressing only one of the tagged proteins to test for non-specific interaction with the resin. For example, a strain with only Msp1-HA should be subjected to an anti-myc pulldown. I appreciate that VDAC could potentially serve as a negative control, but a beta-barrel protein like VDAC could have significantly different propensities for non-specific interaction with resin than the alpha-helical proteins of interest. Another concern with the pulldowns is that overexpression of Msp1 in *S. cerevisiae* is toxic. Is a similar phenotype observed in *T. brucei*?

We have added the three suggested additional negative controls.

1) Cell lines individually expressing TbTsc13-HA, POMP19-HA, POMP31-myc, TbJ31-myc, TbMsp1-myc were subjected to pulldown with anti HA- and myc-beads, respectively. The results show that as expected the tagged proteins were only recovered in the pellet when using resin with matching anti-HA or myc beads. Thus, no unspecific interaction with the resin was observed (see new Fig. S6A).

2) We agree that a beta-barrel protein is not the best marker to monitor unspecific binding. We have now added an additional panel showing that the C-terminally membrane anchored outer membrane protein ATOM69 behaves in the same way as VDAC (see revised Fig. 2).

3) Furthermore, we have added a growth curve showing that expression of myc-tagged TbMsp1 only marginally affects growth *T. brucei*. Using the antiserum against TbMsp1, we show in the same experiment that myc-tagged Msp1 is expressed at similar levels to the endogenous Msp1 which may explain the phenotypic difference in overexpression between yeast and trypanosomes (new Fig. S5).

The new control experiments are now discussed in the revised manuscript (towards the end of the first chapter of the results section).

Another way to validate a stable complex is to use RNAi knockdown. For many stable complexes, the knockdown of one protein can destabilize the whole complex. Does this also occur with the proposed TbMsp1-POMP19-POMP31-TbJ31-TbTsc13 complex? As the inducible RNAi strains are already made (Figure S4), many of the tools required for this experiment are readily available.

We did some of the suggested experiments which required the establishment of novel transgenic cell lines. To that end, the double tagged cell lines (TbTsc13-HA/TbJ31-myc, TbTsc13-HA/POMP31-myc and POMP31-HA/TbJ31-myc) were transfected using the construct allowing for inducible RNAi of TbMsp1. Immunoprecipitation experiments with the resulting TbMsp1-depleted cell lines showed that the complex containing the tested TbMsp1-interacting proteins did not change dramatically, since the pulldown between the pairs of the tagged proteins still worked at least in one direction (see Figure 1 below). However, the interpretation of these results is difficult. It is clear that depletion of TbMsp1 does not result in a global collapse of the OM complex formed by TbMsp1 and tested interacting proteins. To get a more detailed picture would require the production and analysis of many more transgenic cell lines, which we consider beyond the scope of the present study. For this reason, we would not like to add these results to the revised manuscript.

[Figure removed by editorial staff per authors' request]

Since knockdown of pATOM36 affects assembly of ATOM, this can lead to pleiotropic phenotypes and/or indirect effects that will complicate data interpretation. Similarly, knockdown of VCP can also lead to pleiotropic phenotypes. With the pATOM36/VCP double knockdown many factors are changed in the cell and I am very uneasy drawing mechanistic models from simple phenocopy experiments. For example, the authors state "POMP31, TbJ31 and TbTsc13 contribute to TbMsp1 activity in the MAD pathway that is triggered upon pATOM36 depletion", yet the data in Figure 4 lacks sufficient controls to support this statement. This conclusion depends on knockdown of TbJ31, POMP31, TbTsc13

phenocopying Msp1 knockdown in the pATOM36/VCP double knockdown background. I believe that these knockdowns phenocopy Msp1 knockdown, but there are insufficient controls to conclude that they phenocopy is the result of disrupting Msp1 activity. For example, there are no controls to show that this phenotype depends on VCP knockdown, as the model would predict. Likewise, the model predicts that additional knockdown of Msp1 should have no additional effect on ATOM19/46 levels. Unfortunately, neither of these controls were performed.

We agree that there are many possible confounding factors in the interpretation of triple RNAi experiments. We therefore toned down our interpretation and now refer to the activity of the MAD pathway rather than making specific claims regarding Msp1 activity. The specific changes are outlined below:

OLD (Discussion, paragraph 2)

Three of these interactors, POMP31, TbJ31 and TbTsc13 contribute to TbMsp1 activity in the MAD pathway that is triggered upon pATOM36 depletion.

NEW (Discussion, paragraph 3)

Three of these interactors, POMP31, TbJ31 and TbTsc13 contribute to the activity of the MAD pathway that is triggered upon pATOM36 depletion.

Technically we are at the limits with our triple RNAi experiments, Because of a lack of further suitable resistance markers, quadruple RNAi experiments are not possible using our current set up. Adding Msp1-RNAi on top of a triple RNAi cell line, as suggested by the reviewer, is therefore not possible.

In the same vein, the figure title for Figure 4 states "TbJ31, POMP31, and TbTsc13 are required for the degradation of pATOM36 substrates by TbMsp1". The authors do not demonstrate that these proteins are required for Msp1 activity; they only show that knockdown of these proteins phenocopies Msp1 knockdowns. As described above, given the pleiotropic effects of pATOM36 and VCP knockdown, there are many possible explanations for this observation.

We agree and have toned down the phrase in question:

OLD

Figure 4. TbJ31, POMP31 and TbTsc13 are required for the degradation of pATOM36 substrates by TbMsp1

NEW

Figure 4. TbJ31, POMP31 and TbTsc13 are required for the MAD pathway triggered by the absence of pATOM36

In the discussion section the authors state "This degradation requires their selective extraction from the OM by the trypanosomal homologs of the Opisthokont mitochondrial quality control components Msp1 or VCP." The authors have not shown extraction, they have only shown Msp1/VCP dependent degradation in whole cell extracts, not from mitochondria. To show extraction the authors would need to show a change in mitochondrial concentration (integral, not peripheral) that is Msp1/VCP dependent and also show direct interaction of ATOM19 and ATOM46 with Msp1. Demonstrating decreased interaction of Msp1 with ATOM19/46 after knockdown of TbJ31, POMP31, and TbTsc13 could also support the model that these proteins are required for Msp1 activity.

We tried to identify Msp1 substrates using exclusive expression of a substrate trap mutant of TbMsp1, but the experiment did not yield conclusive results. Thus, we agree that we have to tone down the phrase in question:

OLD (Discussion, paragraph 1)

“This degradation requires their selective extraction from the OM by the trypanosomal homologs of the Opisthokont mitochondrial quality control components Msp1 or VCP.

NEW (Discussion, paragraph 1)

“This degradation will require the selective extraction of these membrane proteins from the OM. TbMsp1 and TbVCP, the trypanosomal homologs of the Opisthokont mitochondrial quality control components Msp1 and VCP, are AAA-ATPases and therefore perfect candidates for such a job.”

Furthermore, we also toned down the end of the introduction section:

OLD

Cells depleted of pATOM36 indicate that TbMsp1 removes destabilised OM proteins in concert with cytosolic TbVCP by delivering them to the proteasome. We found that four integral OM proteins stably interact with TbMsp1, and show that, uniquely, three of these interaction partners are required for the full activity of TbMsp1 in this quality control pathway, setting TbMsp1 apart from Opisthokont Msp1.

NEW

It has previously been shown that ablation of pATOM36 triggers a MAD pathway resulting in the proteasomal digestion of destabilised pATOM36 substrates from the OM. Results of the present study, using cells depleted for Msp1 and/or TbVCP, are consistent with the notion that TbVCP and Msp1 contribute to this pathway. In addition, we found four integral OM proteins that interact with TbMsp1, and show that ablation of three of them interferes with the MAD pathway in cells where Msp1 levels are not affected.

TbJ31 expression is significantly decreased upon pATOM36 knockdown (Figure 3C). But further knockdown of TbJ31 rescues the effects of pATOM36 knockdown (Figure 4). Since the authors already have the mass spec data in Figure 3C, they should explicitly say how knockdown of pATOM36 affects the expression levels of POMP31, TbJ31, TbTsc13, POMP19 and Msp1 in the text and in Figure 3C.

POMP31, TbJ31, TbTsc1, POMP19 and Msp1 are all detected in the pATOM36 RNAi SILAC analysis. However, TbJ31 is the only one the level of which is significantly downregulated.

The interactors are now labeled in all panels of the original Fig. 3 and the text has been modified accordingly. Because of the additional labels, we have separated the original Fig. 3ABC into two Figures (NEW Fig. 3AB and NEW Fig. 4), this allowed us to increase the size of the volcano plots in NEW Fig. 4 which makes them more easy to read.

Minor Concerns

- *In Figure 2 with the POMP19-HA and TbMsp1-myc pull down, it appears that there is a non-specific band in the Myc blot. I'm not bothered by the non-specific band, but rather that the crop occurs in the middle of the band. The authors should reformat the figure and specifically label the band as non-specific to facilitate easy data interpretation.*

The cropping problem with Panel1/2nd row was caused by the probing and reprobing of the same immunoblot with two different antibodies detecting proteins of very similar molecular

weight. The experiment has now be repeated with duplicate immunoblots and the panel in question has been replaced (see revised Fig. 2).

- *In Figure S4, why is there is size shift in the EtBr stained rRNAs for POMP19?*

This is a recurring problem that we have seen before. The apparent size shift observed affects all bands of the lane in question. It is most likely caused by residual salt in one of the two isolated RNA fractions used in this analysis.

- *In Figure S5, with the TbMsp1/pATOM36 and TbJ31/pATOM36/TbVCP knockdowns, there appears to be a non-specific band in the pATOM36 blots, but this isn't present in the other blots. Can the authors explain this? I think there are arrows to the bands of interest, but it is very difficult to see this. Bands of interest and non-specific bands should both be clearly labeled.*

We sometimes see non-specific bands when using pATOM36 and TbMsp1 antisera, depending on how any times the diluted antiserum solution has been used. The non-specific bands are indicated in the TbJ31/pATOM36/TbVCP panel. However, in the pATOM36 panel the specific band is indeed very hard to discern. Thus, we have redone the analysis for this cell line and replaced the inset with the new data which gives a much clearer picture (see revised Fig.S8).

- *In Figure S5, there appears to be a non-specific band in the TbMsp1 blot that is not present in Figure S4.*

See above

Reviewer #3 (Comments to the Authors (Required)):

Sustaining normal mitochondrial functions hinges on the presence of sophisticated mitochondrial-associated degradation (MAD) pathways that survey the OMM and monitor its protein precursors for proper targeting and assembly. The highly conserved Msp1, an ATPase associated with diverse cellular activities, plays an instrumental role in this process. In this manuscript the authors report the findings that TbMsp1, the Trypanosoma brucei ortholog of Msp1, is localized to both the mitochondrial OM and glycosomes, a kinetoplastid-specific organelle related to peroxisomes. Further, the authors demonstrated that cells depleted of pATOM36, which mediates complex assembly of a-helically anchored mitochondrial outer membrane proteins, results in TbMsp1 mediating the removal of destabilized OM proteins in concert with cytosolic TbVCP by targeting them to the proteasome. The authors report the identification of four integral OM proteins that stably interact with TbMsp1, and show that, uniquely, three of these interaction partners are required for the full activity of TbMsp1 in this quality control pathway.

Overall, this is an interesting study, and the authors indeed reported a number of important observations. While overall, I would be supportive of publication, there are some points that need be addressed prior to further consideration.

1- The authors show, in their subcellular fractionation of cells expressing TbMsp1-HA and its four epitope-tagged OM interactors, that they all co-fractionate with the voltage dependent anion channel (VDAC) (Fig S2A), as would be expected for mitochondrial OM proteins. It would be informative if a proteinase K-treatment assay (in addition to the carbonate extraction assay shown) is performed on the mitochondrial solubilized fractions to faithfully confirm that all the 4 interacting proteins are predominantly integral membrane proteins.

We are not sure we understood the reviewers argument. The Msp1-interacting proteins are OM proteins and therefore protease-sensitive provided they have a domain that is exposed to the cytosol. In our case the proteins are tagged and the tag likely points to the cytosol, thus it is rapidly removed in the experiment. Thus, while the protease protection experiment can be used to show that the mitos in the digitonin pellet are still intact it only gives very indirect information regarding membrane integration of the proteins.

Nevertheless, we did the proteinase K protection assay and got the expected results. Msp1 and the three interacting proteins which influence the activity of the MAD pathway are sensitive to the proteinase K digestion similar to the OM marker ATOM69, whereas the intermembrane space localized Tim9 and the matrix maker mHsp70 are only digested after the membranes have been dissolved by Triton (new Fig. S2C).

The manuscript has been modified accordingly. A new paragraph describing the findings has been added (third paragraph, in the first chapter of the result section).

2- The authors have demonstrated in figure 1 that TbMsp1 interacts with proteins of the mitochondrial OM and the glycosomes. What is the trigger for this complex formation? Is it constitutive or inducible? Have the authors tried to investigate and monitor the complex formation under the condition of pATOM36 depletion with respect to WT condition? Is the formation of this complex reversible or irreversible? It would be interesting to better clarify these points to faithfully shed the light on the dynamics of this complex formation and to appreciate the context of its potential biological role.

We have repeated the same pulldown experiments in the background of an induced pATOM36 RNAi cell lines. The result shows that the described Msp1-containing mitochondrial complex is present in both the presence and the absence of pATOM36. The result is shown in new Fig. S9 and a paragraph discussing the results has been added at the end of the results section (see below).

“Under wildtype conditions, Msp1 forms a complex with four interacting OM proteins (Fig. 1B), three of which contribute to the activity of the MAD pathway (Fig. 5). We wondered if this complex formed as a response to MAD pathway activation. We performed a SILAC pulldown experiment of the in situ HA-tagged TbMsp1 expressed in induced pATOM36-ablated RNAi cell lines. The bait Msp1 and all four Msp1-interacting OM proteins were found enriched to very similar extents as in wildtype conditions (Fig. S9). Moreover, essentially the same is the case for the glycosomal proteins. Thus, the TbMsp1-containing OM protein complex described in our study is present in both the presence and the absence of pATOM36.”

3- In figure 1B, the authors are advised to increase the figure size and resolution to make the figure a bit clearer for the reader.

The size of Fig. 1B has been increased.

4- In Figure 3, the authors (correctly, I believe) focus on changes in the levels of 2 of 11 OM proteins that are reduced in pATOM36 RNAi. It would be good to have some discussion about the identity and putative functions of the 9 other proteins. Subsequently, they show that several pATOM36 substrates are increased/rescued by RNAi against the proteasome or combined RNAi against MSP1 and VCP. However, in the volcano plots in Figure 3C, there are many additional proteins that appear significantly changed with pATOM36 RNAi (top) or with combined proteasome (middle) or MSP1/VCP (bottom) RNAi. Some analysis (i.e. string or pathway analysis, etc...) and discussion about their function should be provided.

We agree that it makes sense to discuss the results of the pATOM36 RNAi cell line (Fig. 3C, top panel) in some more detail and have added the paragraph below (top page 7, highlighted).

“This group of proteins consists of ATOM subunits, OM membrane proteins of unknown function termed POMP33, TbJ31, VDAC and the putative ABC transporter Tb927.1.4420. Eight of them have been identified as pATOM36 substrates in a previous study (6). Moreover, Tb927.1.4420 and POMP33 were found to be depleted approximately 1.4-fold in the previous study, which is only marginally below the threshold of 1.5-fold. Approximately two thirds of the other proteins found more than 1.5-fold depleted (Fig. 3C, top panel) belong to the mitochondrial importome and thus their depletion likely is an indirect consequence of reduced import due to the diminished levels of the ATOM subunits. However, while the level of the Msp1 interactor TbJ31 was significantly decreased 1.7-fold, the same was not the case for Msp1 itself nor for any of the three remaining interactors. The fact that many more non-OM proteins were detected in the present experiment compared the previous study (6) can be explained because the induction of pATOM36 RNAi was one day longer and because, instead of crude mitochondrial fractions, whole cellular extracts were analysed.”

The topic of the present study is the role TbMsp1 and its interacting proteins play in a novel trypanosomal MAD pathway. The large majority of the non-OM proteins detected in the middle and lower panel of Fig. 3C are non-mitochondrial proteins. This is not surprising since ablation of the beta subunit of the proteasome subunit or of VCP influences a large fraction of the cellular proteome. We therefore believe that doing a string or pathway analysis would be tangential and beyond the scope of the present study.

5- A model figure to recapitulate their findings would be useful.

This is an excellent idea. We prepared such a Figure (see new Fig. 6).

6- There are a few minor typos that needs attention from the authors. For instance, in the introduction, second paragraph, line 4 (plays a key role in in this process). "in" is duplicated.

Corrected.

Reviewer #4 (Comments to the Authors (Required)):

Gerber et al. presented interesting findings about the role of Msp1- containing complex in mitochondrial outer membrane protein quality control. They found that:

- *TbMsp1 interacts with four MOM proteins possibly forming a complex*
- *TbMsp1 involves in degradation of MOM proteins via proteasomal degradation*
- *TbMsp1 and its interactors involves in degradation of MOM proteins via MAD pathway*

*While the referee finds that the findings presented in this study are compelling and potentially would be important to advance knowledge in mitochondrial proteostasis in *T. brucei*, the experimental set ups used in the study are not optimal (see comments below) and might compromise the findings.*

Here are important concerns:

1. The western blot interaction analyses shown in Figure 2A are missing an important control. In the current set up it is impossible to judge whether most of the faint bands detected in the IP fractions were background or interactions. VDAC cannot be used as a control here since different antibodies were used, especially because the different preys were overexpressed. Ideally something like an Ig control should be used here to conclude the dataset.

We are not sure we understood exactly what is meant with Ig control and why, according to this reviewer, VDAC cannot be used as a control. However, Reviewer 1 and 2 also criticized Fig. 2A. We have now added three additional controls for the pull-down data which should help to settle the issues (see below). Moreover, we have analyzed the expression levels of the tagged TbMsp1 and show that it was not overexpressed (see new Fig. S5)

1) Cell lines individually expressing TbTsc13-HA, POMP19-HA, POMP31-myc, TbJ31-myc, TbMsp1-myc were subjected to pull-down with anti HA- and myc-beads, respectively. The results shows that, as expected, the tagged proteins were only recovered in the pellet when using resin with matching anti-HA or myc beads. Thus, there is no unspecific interaction with the resin was observed (see new Fig. S6)

2) We agree that a beta-barrel protein is not the best marker to monitor unspecific binding. We have now added an additional panel showing that the C-terminally membrane anchored outer membrane protein ATOM69 behaves in the same way as VDAC (see revised Fig. 2A).

3) Furthermore, we have added a growth curve showing that expression of myc-tagged TbMsp1 only marginally affects growth *T. brucei*. Using the antiserum against TbMsp1, we show in the same experiment that myc-tagged Msp1 is expressed at the similar levels to the endogenous Msp1 which may explain the phenotypic difference in Msp1 overexpression between yeast and trypanosomes (see new Fig. S5).

The new control experiments are now discussed in the revised manuscript (page 6, highlighted).

2. Cellular protein levels are driven by their synthesis and degradation. To monitor the effect of degradation on protein levels, treatment to shut down protein synthesis, such as with Cycloheximide, is important. Otherwise changes in protein levels can be interpreted as changes in either synthesis or degradation or both. Thus, throughout the manuscript (Fig 3 & 4) such treatment need to be incorporated to conclude any effect on protein degradation. Otherwise the referee is not convinced by the conclusion presented in this study.

We agree that we are measuring steady state level of proteins which are the combined results of protein synthesis and degradation. However, the experiment shown in Fig. 3, using the proteasome beta 1 subunit as a target, addresses this issue. It shows that much of the reduction of the pATOM36 substrate levels in the pATOM36 RNAi cell line is indeed caused by proteasomal degradation. (It should be considered here that RNAi does not completely remove the target protein, thus proteasome activity is reduced but not abolished). Moreover, in a previous publication, we have shown that the addition of the proteasomal inhibitor MG132 completely restores the levels of both ATOM19 and ATOM46 in the induced pATOM36 RNAi cell line (Fig. 1C in (1)).

Nevertheless, we tried the suggested experiment. However, it is quite difficult to do because addition of cycloheximide at the previously established minimal concentration that stops translation quite rapidly kills the cells. Thus, inhibition of translation is possible for maximally two hours, whereas one to three days are needed for RNAi induction. In the experiment shown in the Figure 2 below, we compared ATOM19 levels in the pATOM36 and the pATOM36/TbMsp1 RNAi cell lines before and after RNAi induction (3 days) in the absence and after 1 or 2 hours of incubation with cycloheximide. The results revealed only minor differences in the ATOM19 levels in the presence and absence of cycloheximide.

[Figure removed by editorial staff per authors' request]

3. *The authors offer a model, where TbMsp1 and their interactors are involved in mediating protein extraction from MOM and lead to their degradation by proteasome in the cytosol. To state this, similar experiments to Fig 3 and 4 need to be done in purified mitochondria to see whether depletion of TbMsp1 complex affects extraction of MOM proteins for degradation. Otherwise, the authors need to tone down the speculation that the substrates proteins are actually need to be extracted from MOM.*

We agree, reviewer 2 made the same comment, and have now toned down the phrase in question

OLD

"This degradation requires their selective extraction from the OM by the trypanosomal homologs of the Opisthokont mitochondrial quality control components Msp1 or VCP.

NEW

"This degradation will require the selective extraction of these membrane proteins from the OM. TbMsp1 and TbVCP, the trypanosomal homologs of the Opisthokont mitochondrial quality control components Msp1 and VCP, are AAA-ATPases and therefore perfect candidates for such a job."

The referee feels that these concerns need to be addressed first for this study to be considered as a publication in Life Science Alliance journal.

1. S. Käser *et al.*, Outer membrane protein functions as integrator of protein import and DNA inheritance in mitochondria. *Proc. Natl. Acad. Sci. USA* **113**, E4467-4475 (2016).

July 25, 2023

RE: Life Science Alliance Manuscript #LSA-2023-02004-TR

Prof. André Schneider
University of Bern
Department of Chemistry, Biochemistry and Pharmaceutical Science
Freiestr. 3
Bern 3012
Switzerland

Dear Dr. Schneider,

Thank you for submitting your revised manuscript entitled "A Msp1-containing complex removes orphaned proteins in the mitochondrial outer membrane of *T. brucei*". We would be happy to publish your paper in Life Science Alliance pending final revisions necessary to meet our formatting guidelines.

- please address Reviewer 2's remaining comments
- please add ORCID ID to the secondary corresponding author--they should have received instructions on how to do so
- please use the [10 author names et al.] format in your references (i.e., limit the author names to the first 10)
- please make sure the mass spectrometry data is publicly accessible at this point
- you may want to consider uploading Figure 6 as a Graphical Abstract rather than as a figure, but this is up to you

Figure checks:

- the file labelled "Figure Source Data" is just Figure 2 again, please remove this extra file.
- please provide the Source Data for Figure 5A

A. FINAL FILES:

B. MANUSCRIPT ORGANIZATION AND FORMATTING:

Sincerely,

Reviewer #2 (Comments to the Authors (Required)):

I appreciate the efforts made by the authors to address the concerns raised in the original review. Overall, I find the paper to be a valuable advance to the field, but, as detailed below, I still harbor some concerns. I believe that these concerns can be addressed in the text and I am supportive of publication.

As with the original review, my main concern is with the author's claim about the formation of a stable complex between Msp1, POMP31, TbJ31, TbTsc13, and POMP19. The controls in figure S6 are generally helpful, but there are some issues that limit interpretation. For example, the TbMsp1-Myc blot is very faint compared to the input in Figure 2A. Given the very faint bands of TbMsp1-Myc in the elution fraction for Figure 2A, I do not have sufficient data to judge if the elution of TbMsp1-Myc in the HA-IP is specific or non-specific. There also appears to be some non-specific sticking of the ATOM69 to the anti-Myc resin (I know the prominent lower band is not the right size, but the faint upper band is the correct size). This is visible in the TbTsc13-HA, POMP31-Myc, and POMP19-HA controls. Given the fact that these pulldowns were performed by overexpression and used non-specific SDS elution to monitor interaction between membrane proteins, I still harbor concerns about complex formation.

The fact that RNAi knockdown of TbMsp1 does not destabilize the complex also argues against the formation of a stable complex. I agree with the authors that there are multiple possible explanations for this that go beyond the scope of this paper. I am fine with these results not being included in the final manuscript as the author's suggested.

Ultimately, I do not find the controls convincing for the formation of a stable complex, especially given the fact that AAA proteins such as Msp1 are known to transiently interact with substrates. However, I view the paper as an overall advance for the field. I am willing to support publication if the text is modified to remove any reference to formation of a stable complex.

Reviewer #3 (Comments to the Authors (Required)):

The authors have revised the manuscript according to our comments by performing additional experiments and editing of the text. In my view, this study is now appropriate for publication.

Reviewer #4 (Comments to the Authors (Required)):

I'm satisfied with the authors' responses and have no further concern.

July 31, 2023

RE: Life Science Alliance Manuscript #LSA-2023-02004-TRR

Prof. André Schneider
University of Bern
Department of Chemistry, Biochemistry and Pharmaceutical Science
Freiestr. 3
Bern 3012
Switzerland

Dear Dr. Schneider,

Thank you for submitting your Research Article entitled "A Msp1-containing complex removes orphaned proteins in the mitochondrial outer membrane of *T. brucei*". It is a pleasure to let you know that your manuscript is now accepted for publication in Life Science Alliance. Congratulations on this interesting work.

DISTRIBUTION OF MATERIALS:

Again, congratulations on a very nice paper. I hope you found the review process to be constructive and are pleased with how the manuscript was handled editorially. We look forward to future exciting submissions from your lab.

Sincerely,
